# Periosteal skeletal stem cells can migrate into the bone marrow and support hematopoiesis after injury

Tony Marchand[1,2,3,4]*†, Kemi E Akinnola[3,4]†, Shoichiro Takeishi[3,4], Maria Maryanovich[3,4], Sandra Pinho[3,4,5,6], Julien Saint-Vanne[2], Alexander Birbrair[3,4,7], Thierry Lamy[1,2], Karin Tarte[2,8], Paul Frenette[3,4,5]‡, Kira Gritsman[3,4,5]*

[1]Service d'hématologie Clinique, Centre Hospitalier Universitaire de Rennes, Rennes, France; [2]UMR U1236, INSERM, Université Rennes, EFS Bretagne, Equipe Labellisée Ligue Contre le Cancer, Rennes, France; [3]Ruth L. and David S. Gottesman Institute for Stem Cell and Regenerative Medicine, Albert Einstein College of Medicine, Michael F. Price Center, Bronx, United States; [4]Department of Cell Biology, Albert Einstein College of Medicine, Michael F. Price Center, Bronx, United States; [5]Department of Medical Oncology, Albert Einstein College of Medicine, Bronx, United States; [6]Department of Pharmacology & Regenerative Medicine, University of Illinois at Chicago, Chicago, United States; [7]Department of Dermatology, University of Wisconsin-Madison, Madison, United States; [8]Laboratoire Suivi Immunologique des Thérapeutiques Innovantes, Centre Hospitalier Universitaire de Rennes, Rennes, France

*For correspondence:
tony.marchand@chu-rennes.fr
(TM);
kira.gritsman@einsteinmed.edu
(KG)

†These authors contributed
equally to this work

‡Deceased

Competing interest: The authors
declare that no competing
interests exist.

Reviewing Editor: Rio Sugimura,
The University of Hong Kong,
Hong Kong

## eLife Assessment

The study presents **valuable** insights into the role of periosteal stem cells in bone marrow regeneration. The evidence is **convincing**. The data broadly support their claims and in line with state-of-art methodology. Future study on their model will help to strengthen their discovery further.

**Abstract** Skeletal stem cells (SSCs) have been isolated from various tissues, including periosteum and bone marrow, where they exhibit key functions in bone biology and hematopoiesis, respectively. The role of periosteal SSCs (P-SSCs) in bone regeneration and healing has been extensively studied, but their ability to contribute to the bone marrow stroma is still under debate. In the present study, we characterized a mouse whole bone transplantation model that mimics the initial bone marrow necrosis and fatty infiltration seen after injury. Using this model and a lineage tracing approach, we observed the migration of P-SSCs into the bone marrow after transplantation. Once in the bone marrow, P-SSCs are phenotypically and functionally reprogrammed into bone marrow mesenchymal stem cells (BM-MSCs) that express high levels of hematopoietic stem cell niche factors such as Cxcl12 and Kitl. In addition, using ex vivo and in vivo approaches, we found that P-SSCs are more resistant to acute stress than BM-MSCs. These results highlight the plasticity of P-SSCs and their potential role in bone marrow regeneration after bone marrow injury.

**eLife digest** Bone marrow is the soft tissue inside the bones in our bodies. It is the main production facility for new blood cells and makes billions of blood cells daily. However, like any other tissue or organ, the bone marrow can be damaged, for example, by radiation, chemotherapy drugs, or physical injuries like broken bones.

Bone marrow mesenchymal stromal cells or BM-MSCs are a key component of the bone marrow and are responsible for regulating self-renewal, and for proliferation and differentiation of a group of blood-cell producing stem cells called hematopoietic stem cells or HSCs.

A type of skeletal stem cell, the periosteal skeletal stem cells or P-SSCs, are located in a part of the bone called the periosteum, a thin tissue surrounding long bones. These cells are known to help bones regenerate and heal following a fracture. They are thought to share functional similarities with BM-MSCs. However, it is unclear if P-SSCs also support the recovery of blood cell production after damage to the bone marrow.

To find out more, Marchand, Akkinola et al. used bone tissue derived from genetically engineered mice whose P-SSCs produced a fluorescent tag. Transplanting this bone tissue into otherwise healthy mice mimicked the changes usually seen in the bone marrow after an injury. Microscopy imaging of the transplanted bone at different stages revealed that P-SSCs migrated into the bone marrow after the transplantation injury.

Once in the bone marrow, the P-SSCs developed BM-MSC-like characteristics, producing proteins known to support HSPCs. In other words, P-SSCs effectively transformed into new BM-MSCs, to the extent that the transplanted bone marrow could begin producing blood cells again. Further genetic analysis of P-SSCs and BM-MSCs showed that genes involved in stress resistance were more active in the P-SSCs. This suggests that P-SSCs are better at responding to stress, which may be helpful immediately after an injury.

Marchand, Akkinola et al. have developed a new model to study how the bone marrow repairs itself after it is damaged. These findings may help contribute to a more detailed understanding of the mechanisms behind bone marrow regeneration, as well as treatments to improve recovery following injury.

## Introduction

Bone marrow mesenchymal stem cells (BM-MSCs) are rare self-renewing multipotent stromal cells which are capable of multilineage differentiation into osteoblasts, chondrocytes, and adipocytes (*Caplan, 1991*; *Dominici et al., 2006*; *Frenette et al., 2013*). BM-MSCs are mostly localized around the blood vessels and represent an important component of the hematopoietic stem cell (HSC) microenvironment, also referred to as the niche. BM-MSCs closely interact with HSCs and secrete factors, including the C-X-C motif chemokine ligand (CXCL12) and stem cell factor (SCF), that control their self-renewal, differentiation, and proliferation capacities (*Kunisaki et al., 2013*; *Asada et al., 2017*; *Ding et al., 2012*; *Acar et al., 2015*; *Omatsu et al., 2010*; *Sugiyama et al., 2006*). Several studies have used cell surface markers (CD51[+], PDGFRα[+], Sca-1[+]) or reporter mice (*Lepr*-cre, Nestin (*Nes*)-GFP, *Ng2*-cre) to describe distinct BM-MSC populations with significant overlap (*Kunisaki et al., 2013*; *Asada et al., 2017*; *Ding et al., 2012*; *Méndez-Ferrer et al., 2010*; *Zhou et al., 2014*; *Pinho et al., 2013*; *Morikawa et al., 2009*).

Recent studies have suggested that the periosteum, a thin layer of fibrous material that covers the surface of long bones, is a source of skeletal stem cells (SSCs) for bone regeneration (*Chan et al., 2015*; *Duchamp de Lageneste et al., 2018*; *Debnath et al., 2018*; *Arnsdorf et al., 2009*; *Ortinau et al., 2019*). These periosteum SSCs (P-SSCs) have been described as sharing some characteristics with BM-MSCs (*Chan et al., 2015*; *Duchamp de Lageneste et al., 2018*; *Debnath et al., 2018*). However, the relationship between these different stromal cell populations is poorly understood and the distinction between MSCs and SSCs remains controversial.

While the bone marrow is classically known as a major source of MSCs, the bone cortex represents a richer source of colony-forming units-fibroblasts (CFU-F) (*Pinho et al., 2013*; *Mizoguchi et al., 2014*; *Blashki et al., 2016*). It has been suggested that bone regeneration is mediated by both endochondral and intramembranous ossification and that the periosteum plays an important role in bone

regeneration after injury (*Duchamp de Lageneste et al., 2018*; *Debnath et al., 2018*; *Utvåg et al., 1996*; *Wang et al., 2017b*; *Shapiro, 2008*). However, there is little information on the function of P-SSCs, aside from their crucial role in bone healing and remodeling, and whether they contribute to bone marrow regeneration.

In the present study, we developed and characterized a whole bone transplantation model to study bone marrow regeneration, in which an intact adult femur is transplanted subcutaneously into a recipient mouse (*Picoli et al., 2023*). Shortly after transplantation, bone marrow architecture in the transplanted femur, hereafter referred to as the graft, is severely altered with an expansion of adipocytes that mimics the fatty infiltration classically observed in the bone marrow after chemotherapy or radiation (*Zhou et al., 2017*). This initial destruction of the bone marrow microenvironment is followed by a progressive regeneration of blood vessels and the BM-MSCs network. The graft is progressively colonized by host-derived HSCs, allowing hematopoiesis to resume. Unexpectedly, we found that P-SSCs can migrate into the bone marrow and acquire BM-MSC niche functions, making them capable of supporting hematopoiesis through the in vivo expression of specific niche genes, such as *Cxcl12* and *Kitl*. In addition, we found that BM-MSCs and P-SSCs display different metabolic profiles, and that P-SSCs exhibit higher resistance to transplantation-induced stress. In conclusion, our study demonstrates the high plasticity of P-SSCs and highlights their potential contribution to bone marrow stroma regeneration after injury.

## Results
### The whole bone transplant model recapitulates physiological regeneration of the bone marrow

In order to study the mechanisms involved in bone marrow regeneration, we employed a model system based on the subcutaneous transplantation of an intact adult femur into non-conditioned age- and sex-matched recipient mice (*Figure 1A*). The bone transplantation is followed by a rapid and massive depletion of bone marrow cells in the engrafted femur with a cell viability reaching below 10% at 24 hr after transplantation (*Figure 1—figure supplement 1A*). Notably, bone marrow necrosis following bone transplantation is associated with the replacement of hematopoietic cells in the graft femur by marrow adipocytes, similar to the effects of chemotherapy or irradiation (*Zhou et al., 2017*; *Yamazaki and Allen, 1991*; *Figure 1—figure supplement 1B*). Following bone transplantation, we observed a progressive increase in bone marrow cellularity over time with no significant difference in cellularity between the graft and host femurs at 5 months following transplantation (*Figure 1B*). Regeneration of the bone marrow compartment is associated with a reduction in adipogenic infiltration, as revealed by staining with the anti-perilipin antibody at 4-month post-transplantation (*Figure 1—figure supplement 1B*). Additionally, the absolute number of graft BM-MSCs, defined as CD45$^-$Ter119$^-$CD31$^-$CD51$^+$CD140α$^+$ cells by flow cytometry (*Pinho et al., 2013*), also increased over time, with no difference between the graft and host femurs at 5 months following transplantation (*Figure 1B*). We also observed a progressive increase in the number of Lin$^-$Sca1$^+$cKit$^+$CD48$^-$CD150$^+$ phenotypic HSCs in the graft femur over time (*Figure 1B*). Although the absolute HSC numbers in the graft did not reach HSC numbers as detected in the host femur after 5 months (*Figure 1B*), no differences were observed in the numbers of hematopoietic progenitors or in the frequencies of hematopoietic cell populations between the host and graft femurs (*Figure 1—figure supplement 1C*). These data suggest that bone transplantation causes significant bone marrow injury, followed by a recovery process.

Hematopoiesis is a highly regulated and essential process by which all the differentiated blood cells are produced. To investigate whether functional hematopoietic progenitors fully recover in graft femurs, we performed a non-competitive bone marrow transplantation assay, in which we transplanted either graft or host bone marrow cells into lethally irradiated recipients at 5 months after bone transplantation (*Figure 1C*, *Figure 1—figure supplement 2*). The survival of lethally irradiated recipients remained equal in both groups, with 100% of recipients surviving throughout the length of the experiment. Chimerism analysis revealed robust engraftment of recipient mice with hematopoietic cells derived from the engrafted femurs, indicating that HSCs and progenitors derived from graft femurs can sustain long-term hematopoiesis upon transplantation (*Figure 1D*). We also observed no significant differences in donor cell contribution to myeloid or lymphoid lineages between the two

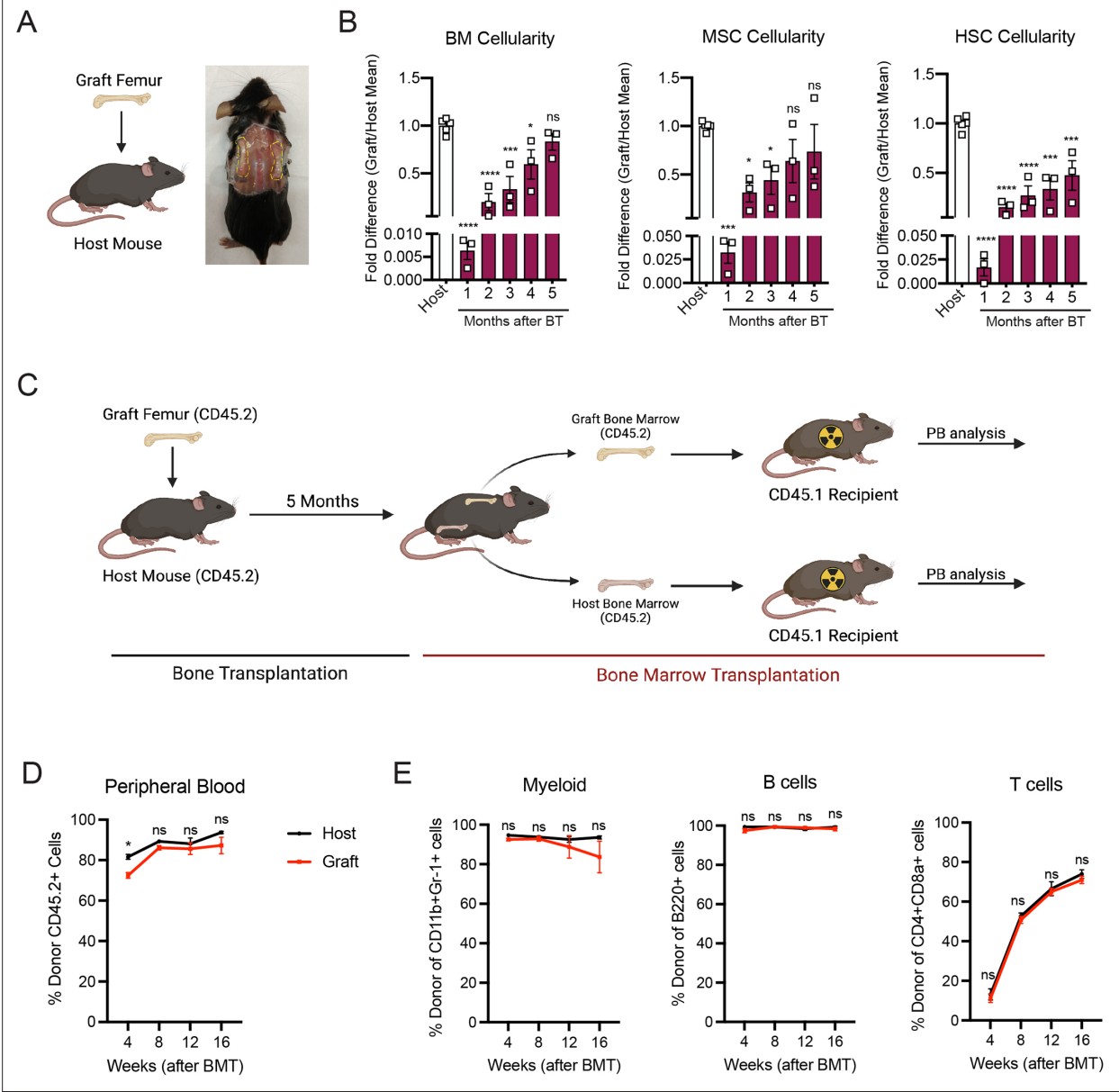

**Figure 1.** Whole bone transplantation is a good model to study bone marrow regeneration. (**A**) Schematic and picture of the bone transplantation procedure. (**B**) Fold difference quantification of graft femur/host femur cellularity normalized to mean host femur cellularity. Total graft bone marrow cells, bone marrow mesenchymal stem cells (BM-MSCs) and hematopoietic stem cells (HSCs) were analyzed monthly until 5 months after bone transplantation (BT) (*n* = 3). Ordinary one-way ANOVA with Dunnett multiple comparisons was used to determine statistical significance. (**C**) Schematic illustration of the non-competitive repopulating assay after bone transplantation. (**D**) Donor HSC contribution of graft and host recipients at 4 weeks after bone marrow transplantation (*n* = 10). (**E**) Quantification of tri-lineage (myeloid, B lymphoid, and T lymphoid cells) engraftment 4-week post-transplantation (*n* = 10). *p < 0.05, **p < 0.01, ***p < 0.001, ****p < 0.0001. This figure was created with BioRender.com.

The online version of this article includes the following figure supplement(s) for figure 1:

**Figure supplement 1.** Bone marrow regeneration after transplantation.

**Figure supplement 2.** Regenerated bone marrow is capable of supporting hematopoietic cells after transplantation.

groups (*Figure 1E*). Altogether, these data establish our bone transplantation model as a useful tool to study bone marrow and HSC-niche regeneration.

## Graft BM-MSCs are graft-derived and progressively express HSC-niche factors during regeneration

Next, we aimed to determine the origin of the hematopoietic and stromal cell populations in the graft bone marrow. To achieve this, we took advantage of the *Rosa26^{mT/mG}*and ubiquitin C (UBC) promoter-driven GFP mouse models, in which all cells are labeled by red and green reporters, respectively (*Muzumdar et al., 2007*; *Liu et al., 2020*). We transplanted femurs isolated from UBC-GFP mice into *Rosa26^{mT/mG}* recipients and quantified endothelial cells, BM-MSCs, and hematopoietic cells in the graft (*Figure 2A, B*). Corroborating the results of *Picoli et al., 2023*, we observed that at 5 months post-transplantation, over 98% of the graft BM-MSCs originated from the graft femur, while over 99% of the hematopoietic cells in the graft originated from the host mouse (*Figure 2C, D*). Interestingly, endothelial cells were derived from both the host and the graft, suggesting the contribution of different progenitors. To further assess endothelial regeneration within the context of bone transplantation, we conducted a transplantation experiment in which femurs from Cdh5 (VE-cadherin)-CreER; iTdTomato (*Wang et al., 2010*) mice were transplanted into WT recipients. Confocal analysis was performed 15 days and 1 month after transplantation (*Figure 2—figure supplement 1A, B*). This revealed the presence of VE-cadherin-labeled vessels within the periosteum and crossing the cortical bone of the graft. This result demonstrates that endothelial regeneration occurs at an early stage following bone transplantation. Furthermore, when femurs from UBC-GFP mice were transplanted into Cdh5-CreER;iTdTomato mice, contributions of both recipient VE-cadherin-labeled and UBC-GFP vessels were observed 5 months later. This result once again demonstrated the dual contribution of endothelial cells to the grafted femur (*Figure 2—figure supplement 1C*). Furthermore, we did not detect any cells derived from the graft in the host femurs (data not shown). These results are consistent with data obtained from ossicle-based experiments, where MSC-seeded ossicles are colonized by recipient-derived hematopoietic cells (*Méndez-Ferrer et al., 2010*; *Sacchetti et al., 2007*; *Friedenstein et al., 1968*; *Tavassoli and Crosby, 1968*; *Varas et al., 2000*).

BM-MSCs are a major constituent of the hematopoietic niche, secreting maintenance factors that support HSCs and hematopoietic progenitors (*Kunisaki et al., 2013*; *Asada et al., 2017*; *Ding et al., 2012*; *Zhou et al., 2014*; *Pinho and Frenette, 2019*; *Greenbaum et al., 2013*). To test for HSC-niche supportive activity of graft BM-MSCs, we utilized *Nes*-GFP reporter mice to isolate BM-MSCs after bone transplantation. Previous work from our group has shown that *Nes*-GFP marks mouse MSCs with HSC-niche function within the bone marrow (*Méndez-Ferrer et al., 2010*). Since our previous analysis (*Figure 2D*) revealed that all of the graft BM-MSCs originate from the graft itself, we transplanted *Nes*-GFP femurs into *Nes*-GFP recipient mice and sorted CD45⁻Ter119⁻CD31⁻*Nes*-GFP⁺ BM-MSCs from donor and recipient mice for analysis at different time points (*Figure 2E*). Since we did not detect circulation of BM-MSCs between host and graft bone marrow in our bone transplantation experiments (data not shown), this strategy allowed us to compare host and graft BM-MSCs using equivalent markers. In addition to the previously described increase over time in the absolute number of BM-MSCs in the graft femur (*Figure 1B*), we observed a progressive increase in the expression of the HSC-niche genes *Cxcl12* and *Kitl* (which encodes Scf), reaching a plateau at 5-month post-transplantation (*Figure 2F*). Similarly, no differences were observed between host and graft femurs in the expression levels of additional niche factors, including *osteopontin* (*Opn*) (*Nilsson et al., 2005*; *Stier et al., 2005*), *angiopoietin-1* (*Angpt1*) (*Arai et al., 2004*), and *vascular cell adhesion molecule-1* (*Vcam1*) (*Jacobsen et al., 1996*; *Papayannopoulou et al., 1998*) at 5 months (*Figure 2—figure supplement 2A*). In this experiment, host and graft BM-MSCs also had similar CFU-F activity at 5 months (*Figure 2—figure supplement 2B*). Altogether, these results show that by 5 months post-bone transplantation, the niche-supportive and ex vivo clonogenic functions of graft BM-MSCs are restored and are similar to the activity of native host BM-MSCs.

## P-SSCs but not BM-MSCs expand early after bone transplantation

SSCs are multipotent cells of the skeletal lineage that are important for bone development, repair, and homeostasis (*Bianco and Gehron Robey, 2000*; *Matsushita et al., 2020*). SSCs have been identified in the periosteum (P-SSCs), compact bone, and bone marrow (*Méndez-Ferrer et al., 2010*; *Zhou*

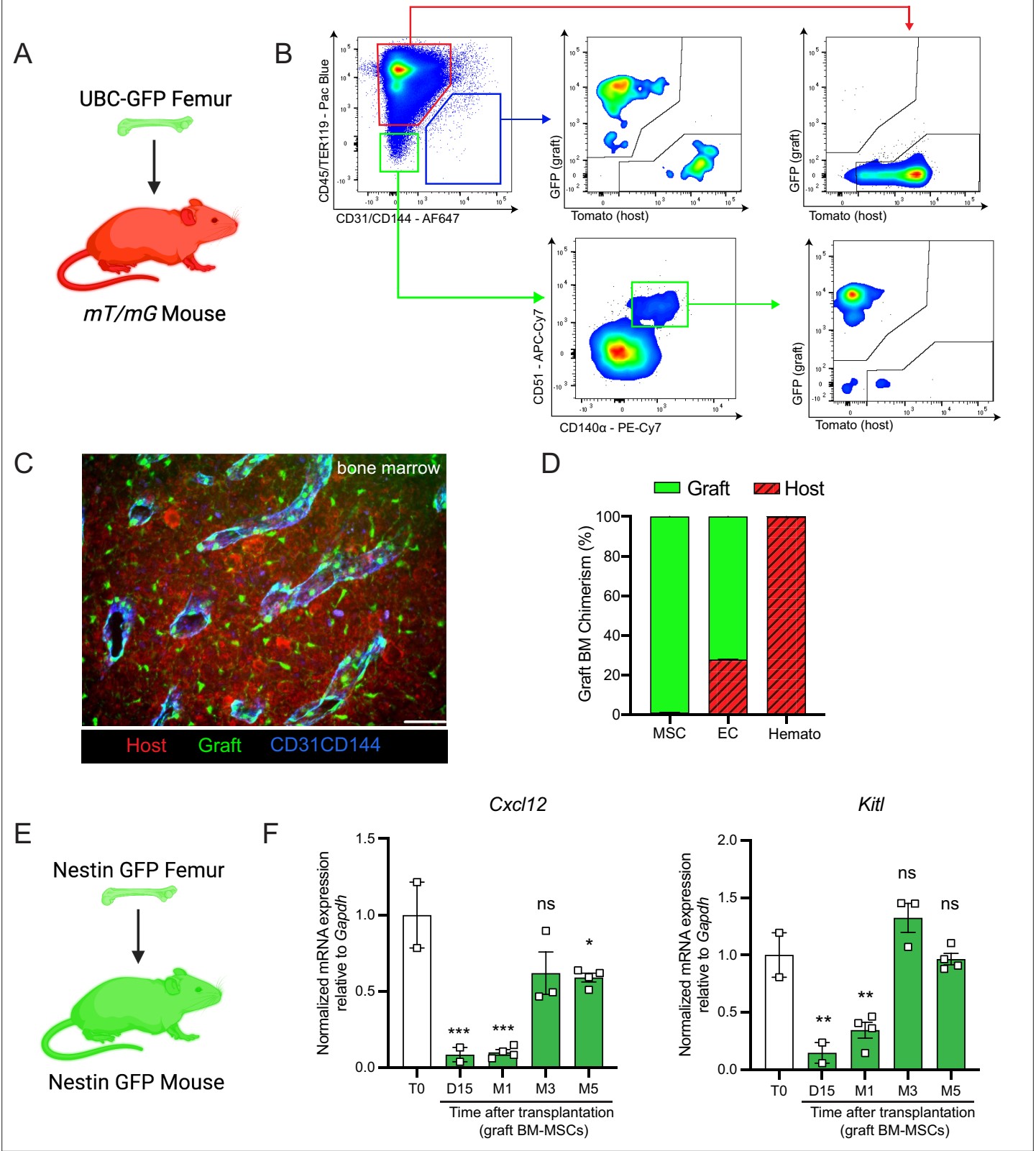

**Figure 2.** Regenerating bone marrow mesenchymal stem cells (BM-MSCs) are graft-derived and express hematopoietic stem cell (HSC)-niche factors. (**A**) Schematic of a UBC-GFP femur transplanted into a *mT/mG* mouse. (**B**) Representative FACS plots showing the gating strategy to determine the origin of the different cell fractions in the graft 5 months after transplantation of a UBC-GFP femur into a *mT/mG* mouse. (**C**) Representative whole-mount confocal z-stack projections of a UBC-GFP bone transplanted into a *mT/mG* recipient 5 months after transplantation. Vascularization was stained

*Figure 2 continued on next page*

*Figure 2 continued*

with anti-CD31 and anti-CD144 antibodies. Scale bar = 100 μm (*n* = 2 mice). (**D**) Origin of graft BM-MSCs, endothelial cells (EC) and hematopoietic cells (Hemato) analyzed by flow cytometry 5 months after bone transplantation (*n* = 2). (**E**) Schematic of the *Nes*-GFP femur transplantation into a *Nes*-GFP mouse recipient. (**F**) Quantitative RT-PCR analysis of mRNA expression of *Cxcl12* and *Kitl* expression relative to *Gapdh* in graft *Nes*-GFP⁺ BM-MSCs compared to steady-state *Nes*-GFP⁺ BM-MSCs at multiple time points after transplantation (*n* = 2–4 mice per time point). One-way ANOVA with Dunnett multiple comparisons was used to determine statistical significance. Data are represented as the mean ± SEM. Unless otherwise noted, statistical significance was determined using unpaired two-tailed Student's *t*-test. *p < 0.05, **p < 0.01, ***p < 0.001,. This figure was created with BioRender.com.

The online version of this article includes the following figure supplement(s) for figure 2:

**Figure supplement 1.** Endothelial regeneration after bone transplantation.

**Figure supplement 2.** GFP+ bone marrow mesenchymal stem cells (BM-MSCs) express hematopoietic stem cell (HSC)-niche factors and form colonies.

*et al., 2014*; *Duchamp de Lageneste et al., 2018*; *Jeffery et al., 2022*). Similar to BM-MSCs, P-SSCs have been shown to have CFU-F activity and the ability to differentiate into osteoblasts, chondrocytes, and adipocytes (*Chan et al., 2015*; *Debnath et al., 2018*; *Mirmalek-Sani et al., 2006*). Due to the severe necrosis and depletion of the marrow cavity content that we observed following bone transplantation (*Figure 1B*, *Figure 1—figure supplement 1A*), we hypothesized that cells derived from the compact bone and/or the periosteum could potentially contribute to stromal marrow regeneration. We first analyzed the cellularity of graft bone marrow, compact bone, and periosteum at different early time points following transplantation. While the number of live cells within the bone marrow and compact bone were drastically reduced in the first 24 hr post-transplantation, we unexpectedly observed a significant but transient increase in live periosteal cells (*Figure 3A*, *Figure 3—figure supplement 1A*). Moreover, while most of the bone marrow cells were depleted shortly after transplantation, cell viability was not affected in the periosteum within this time frame (*Figure 3A*, *Figure 3—figure supplement 1B*). To quantify P-SSCs and BM-MSCs by flow cytometry, we used the combination of CD51 and CD200, as these markers have been well-validated in both tissues (*Chan et al., 2015*; *Debnath et al., 2018*; *Gulati et al., 2018*). Within the CD45⁻Ter119⁻CD31⁻ fraction of live periosteal cells, we confirmed that CD51⁺CD200⁺ P-SSCs had the highest CFU-F activity and were also capable of trilineage differentiation (*Figure 3—figure supplement 1C–E*). Flow cytometric analysis confirmed an expansion of P-SSCs starting at day 3 post-transplantation and peaking at day 8 (*Figure 3B*, *Figure 3—figure supplement 1F*). These results were confirmed by confocal microscopy analysis of *Nes*-GFP graft femurs transplanted into WT mice and stained for Periostin, a matricellular protein highly expressed by periosteal cells (*Duchamp de Lageneste et al., 2018*; *González-González and Alonso, 2018*; *Horiuchi et al., 1999*; *Merle and Garnero, 2012*; *Oshima et al., 2002*). We detected an expansion of *Nes*-GFP⁺ skeletal progenitors (*Tournaire et al., 2020*) within the periosteum with a peak at day 8 (*Figure 3C*), similar to the expansion kinetics that we detected by flow cytometry. Interestingly, by day 15, we could detect Nes-GFP⁺ cells outside of the periosteum layer stained by an anti-periostin antibody and in the compact bone (*Figure 3C*).

To evaluate the potential role of the periosteum in overall bone marrow regeneration, we compared the regenerative capacity of transplanted femurs with intact periosteum to that of femurs in which the periosteum was mechanically removed (*Figure 3D*). At 5 months after transplantation, total cellularity and BM-MSC number were significantly reduced in femurs lacking the periosteum, highlighting a potentially critical role for the periosteum and P-SSCs during bone marrow regeneration.

## P-SSCs are more resistant to stress than BM-MSCs

Due to the ability of P-SSCs to survive bone transplantation, as opposed to BM-MSCs, we explored the intrinsic differences between BM-MSCs and P-SSCs. Using RNA sequencing, we analyzed the transcriptional differences between CD51⁺CD200⁺ P-SSCs and BM-MSCs at steady state. Gene set enrichment analysis (GSEA) revealed that P-SSCs were positively enriched for gene sets associated with stemness and negatively enriched for gene sets associated with proliferation (*Figure 4A*). These results are in line with qPCR analysis showing that P-SSCs express high levels of the cell cycle inhibitor genes *Cdkn1a* and *Cdkn1c*, and low levels of the cell cycle progression gene *Cdk4* at steady state (*Figure 4B*). Additionally, flow cytometric analysis revealed that, compared to BM-MSCs, P-SSCs are less metabolically active, as shown by decreased glucose uptake as assessed by 2-(*N*-(7-nitrobenz-2-oxa-1,3-diazol-4-yl)

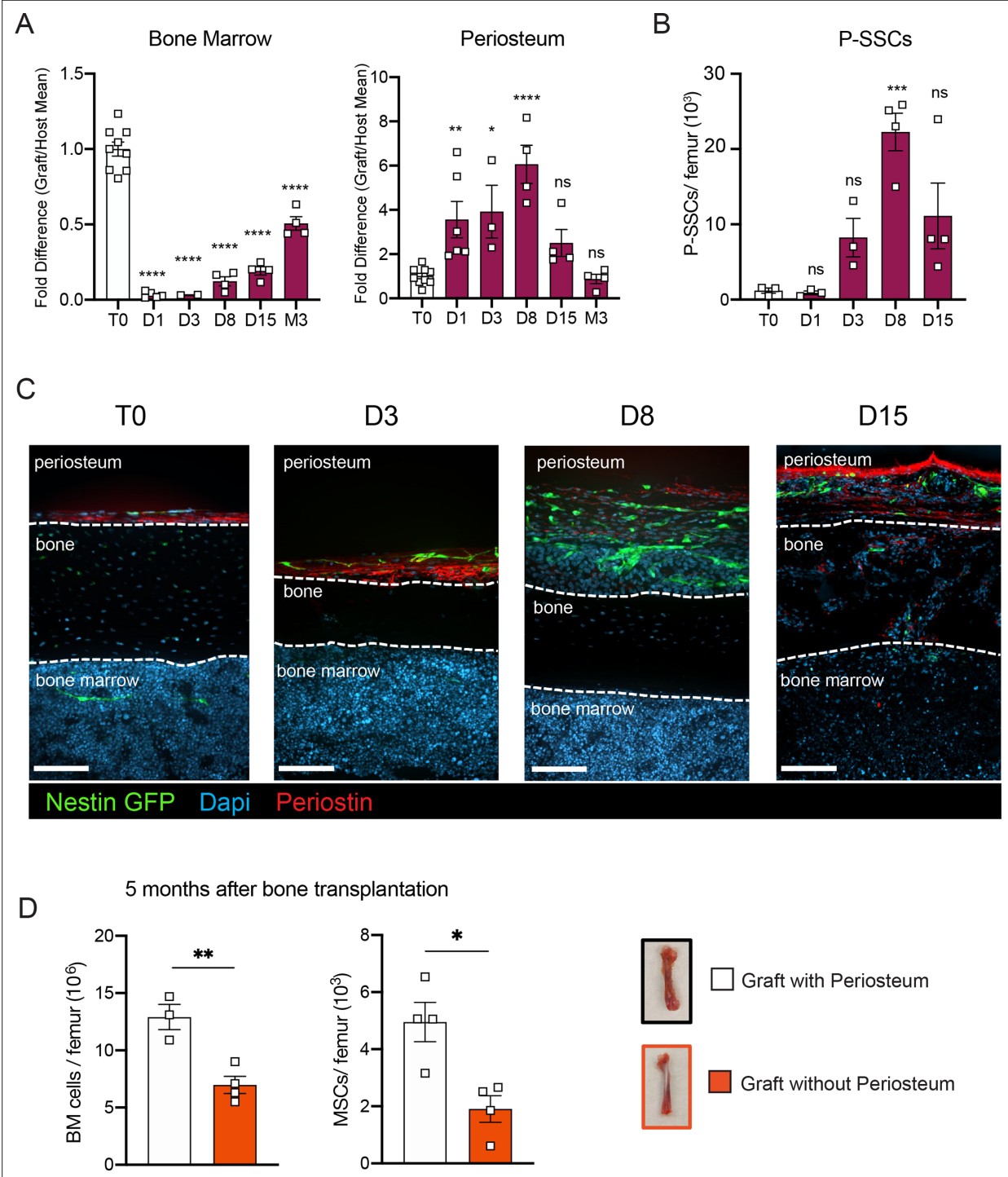

**Figure 3.** P-SSCs remain viable and expand after bone transplantation, in contrast to bone marrow mesenchymal stem cells (BM-MSCs). (**A**) Flow cytometric quantification of fold difference of total graft bone marrow and periosteum cellularity to total steady-state cellularity. Different time points early after transplantation were analyzed (*n* = 2–8). One-way ANOVA with Dunnett multiple comparisons was used to determine statistical significance. (**B**) Absolute number of CD45⁻Ter119⁻CD31⁻CD51⁺CD200⁺ P-SSCs at steady state and 1-, 8-, and 15-day post-transplantation (*n* = 3–4 mice per time point). One-way ANOVA with Dunnett multiple comparisons was used to determine statistical significance. (**C**) Representative whole-mount confocal z-stack projections of *Nes*-GFP⁺ bone graft at steady state, 3-, 8-, and 15-day post-transplantation. Three independent experiments yielded similar results. Arrowheads point at *Nes*-GFP⁺ cells within the bone and bone marrow. Scale bar = 100 µm. (**D**) Total bone marrow cellularity and BM-MSC absolute number 5 months after transplantation of bones with or without intact periosteum (*n* = 3–4 mice per group). Data are represented as the mean ± SEM.

*Figure 3 continued on next page*

*Figure 3 continued*

Unless otherwise noted, statistical significance was determined using unpaired two-tailed Student's *t*-test. *p < 0.05, **p < 0.01, ***p < 0.001, ****p < 0.0001.

The online version of this article includes the following figure supplement(s) for figure 3:

**Figure supplement 1.** Periosteal stem cell expansion after transplantation.

amino)-2-deoxyglucose (2-NDBG) (*Figure 4C*). This led us to hypothesize that P-SSCs are more resistant to stress than BM-MSCs.

These observations led us to also use RNA sequencing to compare steady-state P-SSCs to P-SSCs at 3 days after bone transplantation. We chose the 72 hr time point to capture the time when P-SSCs are expanding but have not yet hit the peak of expansion at around day 8. We found that while steady-state P-SSCs and day 3 graft P-SSCs are quite distinct from steady-state BM-MSCs, there is a clear variance starting to occur between the two P-SSC populations (*Figure 4—figure supplement 1A*). As early as 3 days post bone transplantation, we observed a downregulation of extracellular matrix (ECM) factors such as fibromodulin (*Fmod*) and nidogen 1 (*Nid1*) in the graft P-SCCs (*Figure 4—figure supplement 1B*). Additionally, GSEA comparing steady-state P-SSCs with graft P-SSCs, revealed an upregulation of cell cycle and DNA replication signatures and downregulation of ECM–receptor interaction and glutathione metabolism (*Figure 4—figure supplement 1C*), indicating that a change in the P-SSCs phenotype occurs early after bone transplantation.

Low levels of reactive oxygen species (ROS) and high expression of antioxidant enzymes are mechanisms that help stem cells to avoid stress-induced cell death (*Ito and Suda, 2014*; *Suda et al., 2011*). Thus, we measured ROS levels by staining cells with the superoxide indicator dihydroethidium (DHE). At steady state, flow cytometric analysis revealed that P-SSCs had lower levels of cellular ROS than BM-MSCs (*Figure 4D*). Under physiological conditions, cells can maintain low ROS levels by expressing antioxidant enzymes. Indeed, qPCR analysis revealed higher expression of three major ROS-detoxifying enzymes genes: superoxide dismutase (*Sod1*), glutaminase (*Gls*) and glutathione peroxidase (*Gpx1*), in sorted CD51$^+$CD200$^+$ P-SSCs than in CD51$^+$CD200$^+$ BM-MSCs (*Figure 4E*). Altogether, these results suggest that P-SSCs are more stress-resistant than BM-MSCs, which may leave P-SSCs poised to proliferate in response to transplantation.

The location of the periosteum on the exterior of the bone, exposing P-SSCs to a higher oxygen tension than BM-MSCs, may be a contributing factor to the observed differences in resilience between P-SSCs and BM-MSCs. Therefore, to determine whether the anatomic location of BM-MSCs and P-SSCs is the primary determinant of their differential stress response, we performed an ex vivo culture experiment designed to subject BM-MSCs and P-SSCs to equivalent levels of stress while in the same environment. After short-term ex vivo expansion of total bone marrow and periosteal cells, we performed a lineage depletion of CD45$^+$ hematopoietic cells in both fractions. Purified P-SSCs and BM-MSCs were then maintained for 12 hr in serum-free culture media to stress the cells and mimic the nutrient deprivation that occurs immediately following bone transplantation (*Figure 4F*). Flow cytometric analysis of activated caspase-3/7 revealed that after 12 hr in serum-free media, BM-MSCs exhibited a significantly higher level of apoptosis compared to P-SSCs (*Figure 4G*, *Figure 4—figure supplement 2*). These results suggest that P-SSCs are more intrinsically resilient than BM-MSCs, even when they are subjected to similar stress conditions in an equivalent environment.

## P-SSCs as a source of functional BM-MSCs during regeneration

As we observed that bone transplantation was followed by a depletion of bone marrow cellularity with an early expansion of P-SSCs, and that transplantation of bones without periosteum negatively impacts graft regeneration (*Figure 3D*), we hypothesized that proliferating P-SSCs migrate into the bone marrow and contribute to stromal regeneration. To test this hypothesis, we removed the periosteum from WT femurs and wrapped these femurs with the periosteum isolated from UBC-GFP mice (*Figure 5A, B*). We then transplanted these bones into WT host mice and observed GFP$^+$ cells within the compact bone at 5 months after transplantation (*Figure 5C*), consistent with the well-described role of the periosteum in bone remodeling (*Duchamp de Lageneste et al., 2018*; *Debnath et al., 2018*; *Julien et al., 2020*). Consistent with our hypothesis, we also observed GFP$^+$ cells enwrapping

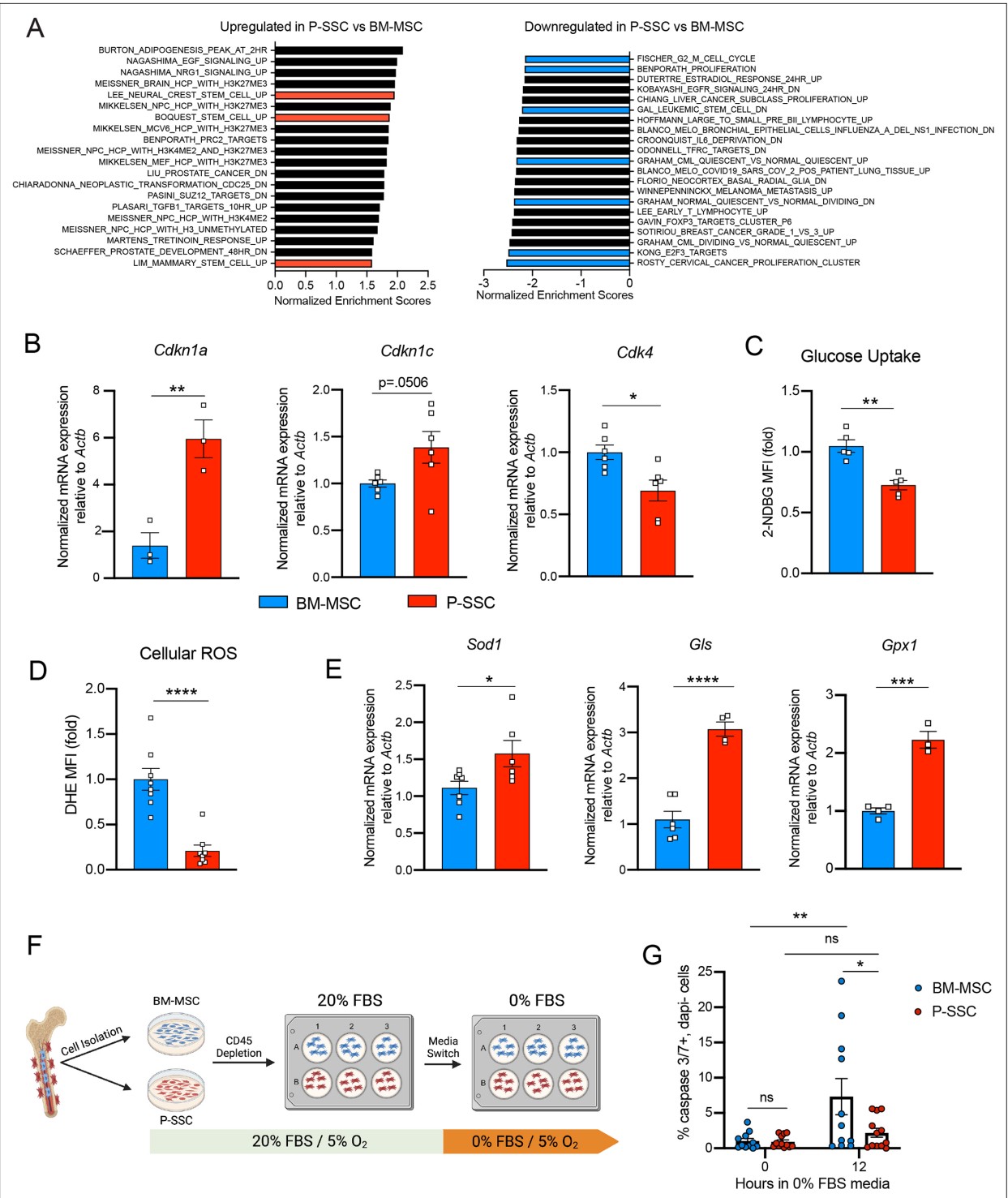

**Figure 4.** Periosteal skeletal stem cells (SSCs) have a metabolic profile conferring a resistance to stress. (**A**) Gene set enrichment analysis (GSEA) plots comparing P-SSCs versus bone marrow mesenchymal stem cells (BM-MSCs) at steady state (*n* = 3 per group). (**B**) Quantitative RT-PCR analysis of mRNA expression of *Cdkn1a*, *Cdkn1c*, *Cdk4* relative to *Actb* in sorted CD45⁻Ter119⁻CD31⁻CD51⁺CD200⁺ BM-MSCs and P-SSCs (*n* = 3–6 per group). (**C**) Flow cytometric analysis of glucose uptake at steady state in CD45⁻Ter119⁻CD31⁻CD51⁺CD200⁺ BM-MSCs and P-SSCs (*n* = 5 per group). (**D**) Quantification of cellular reactive oxygen species (ROS) at steady state in CD45⁻Ter119⁻CD31⁻CD51⁺CD200⁺ BM-MSCs and P-SSCs (*n* = 8 per group). (**E**) Quantitative RT-PCR analysis of mRNA expression of *Sod1*, *Gls*, and *Gpx1* relative to *Actb* in sorted CD45⁻Ter119⁻CD31⁻CD51⁺CD200⁺ BM-MSCs and P-SSCs (*n* = 3–7 per group). (**F**) Schematic illustration of the protocol for the in vitro apoptosis assay. BM-MSCs and P-SSCs were isolated and digested before plating in a 10-cm dish. At near confluence, cells underwent CD45 lineage depletion and plated into multi-well plates. At near confluence, medium was

*Figure 4 continued on next page*

*Figure 4 continued*

switched from 20% to 0% fetal bovine serum (FBS). Cells were analyzed at the time of medium switch and 12 hr. (**G**) Percentage of apoptotic BM-MSCs and P-SSCs cultured under 5% $O_2$ at baseline and 12 hr after being in 0% FBS serum conditions (*n* = 11–12 per group). Two-way ANOVA with Tukey's multiple comparisons test was used to determine statistical significance. Data are represented as the mean ± SEM. Unless otherwise noted, statistical significance was determined using unpaired two-tailed Student's *t*-test. *p < 0.05, **p < 0.01, ***p < 0.001, ****p < 0.0001. This figure was created with BioRender.com.

The online version of this article includes the following figure supplement(s) for figure 4:

**Figure supplement 1.** RNA sequencing analysis of periosteal skeletal stem cells (SSCs), 3-day graft periosteal and SSCs, and bone marrow MSCs.

**Figure supplement 2.** Comparison of periosteal skeletal stem cells (SSCs) and bone marrow MSCs.

endomucin-stained sinusoids and forming a network, similar to the perivascular nature of BM-MSCs (*Sugiyama et al., 2006*; *Méndez-Ferrer et al., 2010*; *Figure 5C*).

Flow cytometric analysis of the graft at 5-month post-transplantation confirmed the presence of periosteum-derived GFP+ MSCs within the bone marrow cavity (*Figure 5—figure supplement 1A*). Importantly, while P-SSCs do not express *Cxcl12* or *Kitl* at steady state, periosteum-derived GFP+ BM-MSCs expressed these niche cytokines at a similar level to control sorted *Nes*-GFP+BM-MSCs (*Figure 5D*). We also quantified the expression of the niche factors *Angpt1* and *Opn*. Similarly, we found that *Angpt1* expression in the graft GFP+ BM-MSCs reached the level of control BM-MSCs, while *Opn* expression was higher in GFP+ BM-MSCs than in control *Nes*-GFP+ BM-MSCs at 5 months after transplantation (*Figure 5—figure supplement 1B*). However, *Opn* has been shown to be upregulated in settings of inflammation, injury, and migration (*Denhardt et al., 2003*; *Wang et al., 2017a*; *Zou et al., 2013*). Therefore, the moderate increase in *Opn* expression level in graft BM-MSCs compared to steady-state BM-MSCs could be due to a residual inflammatory effect of bone transplantation. These results show that P-SSCs can both migrate into the bone marrow cavity and upregulate HSC maintenance genes to support hematopoiesis.

To confirm these results, we took advantage of a previously described transgenic mouse model in which an inducible Cre is placed under the promoter of the *Periostin* gene (*Postn*^MCM^), hereafter referred to as *Postn*^Cre-ER^ (*Kanisicak et al., 2016*). *Postn* encodes the secreted matricellular protein Periostin, and is highly expressed by periosteal cells and upregulated during bone healing and formation (*Duchamp de Lageneste et al., 2018*). While *Postn* is expressed by multiple cell types, including but not limited to osteoblasts and fibroblasts (*Horiuchi et al., 1999*; *Oshima et al., 2002*; *Oka et al., 2007*), its high expression in P-SSCs compared with BM-MSCs (*Figure 6—figure supplement 1A*) makes it a useful marker to distinguish endogenous BM-MSCs from periosteum-derived BM-MSCs after bone transplantation. We crossed the *Postn*^Cre-ER^ line with *Rosa26*^LSL-tdTomato^ reporter (Tomato) mice to be able to lineage trace P-SSCs. We transplanted femurs from *Postn*^Cre-ER^;tdTomato mice into WT CD45.2 recipient mice, and then injected the host mice with tamoxifen shortly after transplantation to induce Cre recombination and Tomato expression in periosteal cells (*Figure 6A*). At early time points following bone transplantation, we see Tomato+ cells confined to the periosteum and located perivascularly. At 21 days after transplantation, we observe Tomato+ cells migrating into the bone marrow (*Figure 6B*). By 5 months after transplantation, we observe robust Tomato+ labeling in the bone marrow located around the vasculature by confocal imaging (*Figure 6—figure supplement 1B*), consistent with our prior observations (*Figure 5C*). Additionally, flow cytometric analysis revealed that an average of 85.4% (range: 65.0–94.5%) of the BM-MSCs within the engrafted bone marrow were Tomato+, indicating a periosteal origin (*Figure 6C*, *Figure 6—figure supplement 1C*).

We then wanted to assess our *Postn* mouse model using irradiation as a more physiologically relevant model of bone marrow injury, and to compare the effects of localized irradiation to femur transplantation on P-SSC plasticity. Single hindlimbs of *Postn*^Cre-ER^;tdTomato mice were irradiated in order to compare the effects of irradiation within the same mouse (*Figure 6—figure supplement 2A*). While a clear difference in the architecture of bone marrow vasculature was observed between the non-irradiated and irradiated limbs, no difference in P-SSCs expansion or migration into the bone marrow was detected, contrasting with the findings in our femur transplant model (*Figure 6—figure supplement 2B*). This observation suggests that bone transplantation results in a more severe injury to the bone marrow compared to localized irradiation, thereby enabling the assessment of BM-MSCs' recovery after more severe injury. However, it is also conceivable that stronger irradiation or a longer observation period would be necessary to observe a similar phenotype.

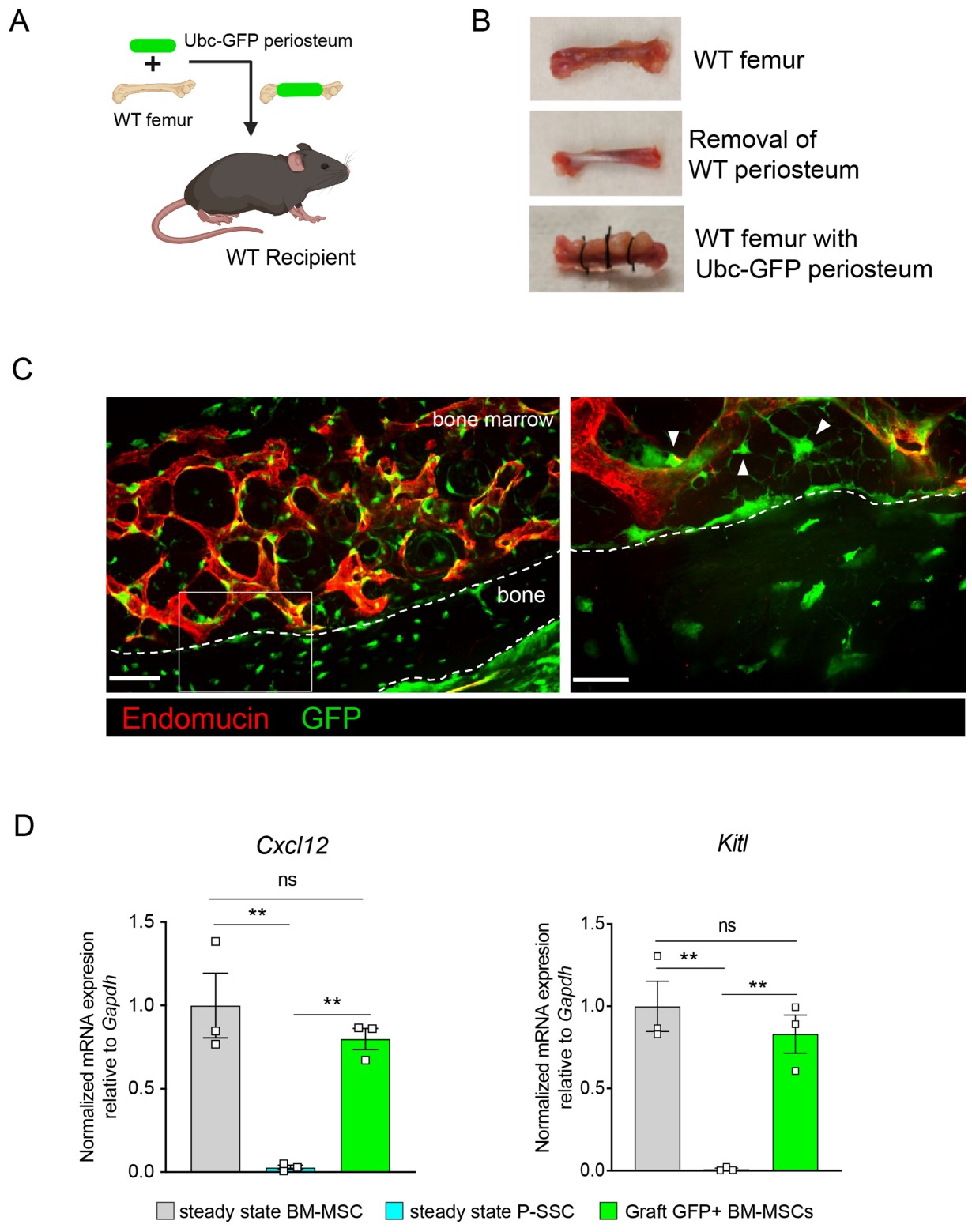

**Figure 5.** Periosteal skeletal stem cells (SSCs) migrate into the bone marrow and support stromal regeneration after bone transplantation. (**A**) Schematic of the transplantation of a WT bone enwrapped with periosteum from a UBC-GFP mouse donor into a WT recipient mouse. (**B**) Pictures illustrating the transplantation of a WT bone enwrapped with periosteum from a UBC-GFP mouse donor into a WT recipient mouse. (**C**) Representative whole-mount confocal z-stack projections of wild-type bone graft enwrapped with periosteum from a UBC-GFP mouse donor into a WT recipient mouse 5 months after transplantation. Three independent experiments yielded similar results. Right panel: arrows pointing to GFP+ periosteum located

*Figure 5 continued on next page*

*Figure 5 continued*

perivascularly. Scale bar = 50 µm (left panel) and 20 µm (right panel). (**D**) Quantification of *Cxcl12* and *Kitl* mRNA levels relative to *Gapdh* in sorted control CD45⁻Ter119⁻CD31⁻Nestin-GFP⁺ bone marrow mesenchymal stem cells (BM-MSCs), CD45⁻Ter119⁻CD31⁻CD51⁺CD200⁺ P-SSCs, and CD45⁻Ter119⁻CD31⁻CD51⁺CD200⁺GFP⁺ periosteum-derived graft BM-MSCs (*n* = 3–4 per group). One-way ANOVA with Tukey's multiple comparisons was used to determine statistical significance. Data are represented as the mean ± SEM. Unless otherwise noted, statistical significance was determined using unpaired two-tailed Student's *t*-test. \*\*p < 0.01. This figure was created with BioRender.com.

The online version of this article includes the following figure supplement(s) for figure 5:

**Figure supplement 1.** Periosteal skeletal stem cells (SSCs) migrate into the bone marrow and upregulate hematopoietic stem cell (HSC)-niche factors.

To examine changes in periosteum-derived BM-MSCs at the gene expression level, we performed bulk RNA sequencing on sorted CD51⁺CD200⁺Tomato+ BM-MSCs from graft *Postn^Cre-ER*;tdTomato femurs at 5 months after transplantation and on sorted steady-state CD51⁺CD200⁺ BM-MSCs and P-SSCs. Venn diagram and principal component analysis revealed that periosteum-derived graft BM-MSCs display a gene expression profile distinct from that of both steady-state BM-MSCs and steady-state P-SSCs (*Figure 6—figure supplement 3A, B*). Consistent with our previous tracing experiment using UBC-GFP periosteum (*Figure 5D*), we observed an upregulation of HSC-niche-associated maintenance genes in the 5-month-old graft BM-MSCs compared to P-SSCs at steady state (*Figure 6D, E*). We also observed the downregulation of *Postn* and other extracellular matrix-related genes, such as fibronectin (*Fn1*) and fibromodulin (*Fmod*), in periosteum-derived graft BM-MSCs compared with P-SSCs at steady state (*Figure 6D, E*). Taken together, our results indicate that P-SSCs can be reprogrammed and adapt a niche-supportive phenotype akin to native BM-MSCs after migrating into the bone marrow following acute stress and subsequent regeneration.

## Discussion

Bone marrow regeneration is a critical process that enables the recovery of hematopoiesis after injury such as irradiation or chemotherapy. Bone marrow mesenchymal stromal cells represent a key component of the bone marrow microenvironment. These stromal cells play a crucial role in regulating the self-renewal, differentiation, and proliferative properties of HSCs. The periosteum, a thin membrane that is highly vascularized and innervated, is located on the outside of the bone. This membrane contains numerous skeletal stromal cells, which play a pivotal role in maintaining the bone tissue and facilitating post-fracture healing. While the bone marrow microenvironment at steady state has been extensively studied, the mechanisms of bone marrow regeneration and stromal recovery remain poorly understood. Furthermore, data related to the functions of P-SSC beyond their established role in bone maintenance and fracture healing remain scarce. The objective of this study was to develop and characterize a model of whole bone transplantation in order to study bone marrow regeneration in mice. In this model, severe injury to hematopoietic and stromal cells within the bone marrow is induced, allowing for the investigation of the regeneration process of both cell populations. We found that the total graft bone marrow cellularity and BM-MSC cellularity demonstrate a gradual increase over time, ultimately reaching levels comparable to steady-state levels by 5-month post-transplantation. The 5-month time point was subsequently employed for further analyses. Prior research has emphasized the significance of stromal integrity for the recovery of HSCs following irradiation or chemotherapy (*Banfi et al., 2001*; *Lucas et al., 2013*; *Hérodin and Drouet, 2005*). The findings of our study indicate that, while initially impacted by the stress associated with transplantation, BM-MSCs ultimately demonstrate their capacity to regenerate and sustain hematopoiesis within the engrafted femur. Furthermore, our findings indicate that hematopoietic progenitors derived from the graft femur are capable of engraftment in secondary recipient mice, thereby facilitating multilineage reconstitution. This model system recapitulates a recovering bone marrow microenvironment. Furthermore, it allows for genetic in vivo analysis of bone marrow regeneration. Consequently, bone transplantation can be employed as a valuable tool in future studies for investigating bone marrow regeneration.

The origin of endothelial progenitors in the bone marrow is not well defined. A recent study showed that during bone marrow regeneration after chemotherapy, sinusoidal and arteriolar vessels are derived from distinct progenitors (*Xu et al., 2018*). Intriguingly, in the present study, we observed that endothelial cells within the graft originate from both the graft and host, which supports the

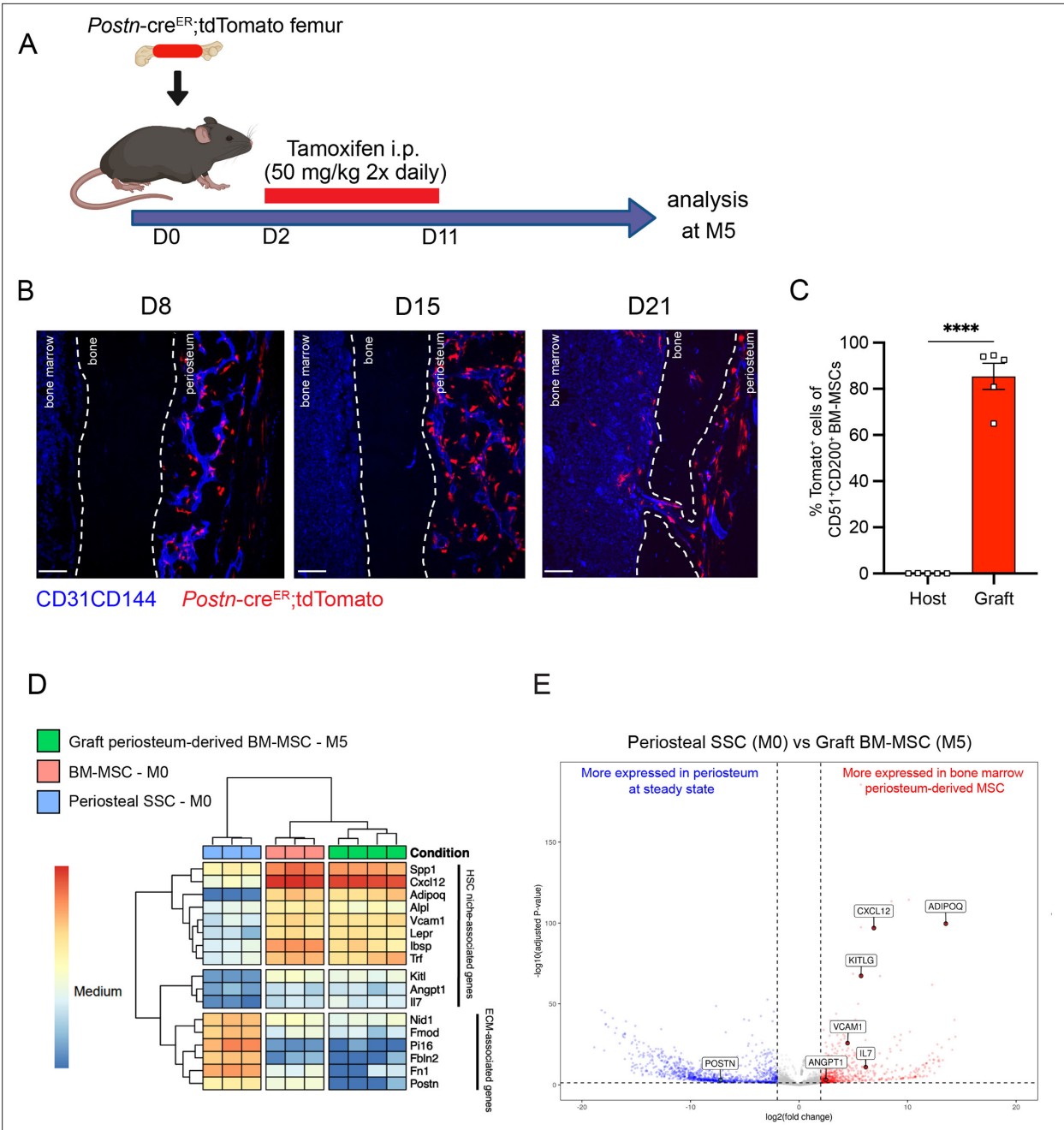

**Figure 6.** Periosteum-derived graft bone marrow mesenchymal stem cells (BM-MSCs) adopt characteristics of baseline BM-MSCs, including the expression of hematopoietic stem cell (HSC)-niche factors. (**A**) Schematic illustration of the transplantation of a *Postn*[Cre-ER];tdTomato femur into a WT recipient mouse. (**B**) Representative whole-mount confocal z-stack projections of transplanted *Postn*[Cre-ER];tdTomato femurs into a WT recipient 8-, 15-, and 21 days after transplantation. Two to three independent experiments yielded similar results. Scale bar = 100 μm. (**C**) Percentage of graft periosteum-derived BM-MSCs labeled Tomato+ 5 months after transplantation of a bone from a *Postn*[Cre-ER];tdTomato mouse into a WT recipient (*n* = 5). (**D**) Heatmap expression level of selected genes defined by previous studies for HSC-niche cells and extracellular matrix genes (*n* = 3–4). (**E**) Volcano plot of P-SSCs compared to graft BM-MSCs showing higher expression of HSC-niche-associated genes in graft BM-MSCs. Data are represented as the mean ± SEM. Unless otherwise noted, statistical significance was determined using unpaired two-tailed Student's *t*-test. ****p < 0.0001. This figure was created with BioRender.com.

The online version of this article includes the following figure supplement(s) for figure 6:

**Figure supplement 1.** *Postn*-Cre[ER];tdTomato can be used to lineage trace periosteal skeletal stem cells (SSCs) and label periosteum-derived bone marrow mesenchymal stem cells (BM-MSCs).

**Figure supplement 2.** Single limb irradiation of *Postn*[Cre-ER];tdTomato mice.

*Figure 6 continued on next page*

*Figure 6 continued*

**Figure supplement 3.** Graft Tomato⁺ bone marrow mesenchymal stem cells (BM-MSCs) derived from P-SSCs exhibit profile similar to that of steady-state BM-MSCs.

hypothesis that different progenitors contribute to the bone marrow vascular network. Therefore, it is likely that endothelial regeneration in the graft, and vascular anastomoses between the host and graft femur are also important for the hematopoietic regeneration process. Further studies are needed to clarify the respective contributions of graft- and host-derived endothelial progenitors and the relative contributions of these different progenitor populations to regeneration of the vascular network and to hematopoietic regeneration.

Our results also highlight the high resilience and plasticity of P-SSCs and reveal their potential contribution to the bone marrow stromal network and bone marrow regeneration. At steady state, P-SSC do not express HSC maintenance genes, such as *Kitl* and *Cxcl12*, and the potential capacity of P-SSC to support HSCs has not been previously addressed. Unexpectedly, our results show that in our model, P-SSCs can migrate to the bone marrow and adopt a phenotype similar to that of BM-MSCs. Therefore, it is possible that P-SSCs can be harvested and manipulated as a source of BM-MSCs. Interestingly, we did not observe a similar migration and plasticity of P-SSCs in the context of localized femur irradiation. This observation suggests that femur transplantation introduces a greater stress on the bone marrow than irradiation, and that this increased level of stress is required to induce P-SSC plasticity and migration into the bone marrow.

In accordance with our findings, a recent study employing Gli1 to trace P-SSC demonstrated the localized expression of *Kitl* and *Cxcl12* by P-SSCs at the fracture site (*Jeffery et al., 2022*). However, this model was not designed to specifically address the role of P-SSC in bone marrow regeneration. Furthermore, niche-specific genes were only expressed by cells adjacent to the fracture callus. Although we did not find a baseline difference in *Gli1* expression between BM-MSCs and P-SSCs in our RNA sequencing analysis, this may be due to the differences in surface markers and reporters used to identify P-SSCs and BM-MSCs. In the same study, authors did not find expression of reporters in the periosteum using the *Postn*^Cre-ER mice. It is likely that this discrepancy is attributable to differences in generation of the Postn-Cre mice used. Indeed, previous studies have demonstrated that, compared to steady state, the *Postn* gene is upregulated following the activation of P-SSC, which is clearly evident in the context of whole bone transplantation (*Duchamp de Lageneste et al., 2018*). The results of our flow cytometry and imaging analyses demonstrate that P-SSC is specifically labeled at early time points following the transplantation of grafts from *Postn*^Cre-ER mice. Therefore, we show a novel application for the use of the inducible *Postn*^Cre-ER mice to differentiate between BM-MSCs and P-SSCs in vivo. Accordingly, Duchamp de Lageneste et al. demonstrated that periostin contributes to the highly regenerative nature of P-SSCs compared to BM-MSCs (*Duchamp de Lageneste et al., 2018*). While previous studies have used *Prx1* and *Ctsk*-Cre models, these models do not adequately allow the distinction between P-SSCs and BM-MSCs, likely due to their common embryonic origin (*Duchamp de Lageneste et al., 2018*; *Debnath et al., 2018*; *Bianco and Gehron Robey, 2000*). Periostin is a well-studied protein that has been shown to interact with extracellular matrix proteins and plays a key role in tissue regeneration and cancer progression, promoting proliferation, invasion, and anti-apoptotic signaling (*González-González and Alonso, 2018*; *Bao et al., 2004*; *Butcher et al., 2007*; *Conway et al., 2014*). Therefore, it is possible that periostin contributes to P-SSC proliferation and migration into the bone marrow, which would be an interesting area for future investigation.

Additionally, our findings illustrate the differential stress response between BM-MSCs and P-SSCs. While BM-MSCs are renowned for their resilience to stress, our findings illustrate that P-SSCs exhibit an even greater resistance to stress, which is attributed, at least in part, to their distinctive metabolic profile (*Morikawa et al., 2009*; *Haas et al., 1969*). Given that differences in apoptosis were observed between BM-MSCs and P-SSCs, even when they were cultured ex vivo under identical stress culture conditions with identical oxygen tension, it can be concluded that the observed differences between P-SSCs and BM-MSCs are due to intrinsic cellular properties rather than their anatomical location. Nevertheless, additional studies are required to elucidate the underlying mechanism responsible for the observed relative stress resistance of P-SSCs.

In conclusion, we have used a whole bone transplantation model to study bone marrow regeneration in vivo in response to acute injury using genetic tools. Our study has shown that through their

high level of resilience and plasticity, P-SSCs can facilitate BM-MSC regeneration. Our data suggest that P-SSCs are able to contribute to hematopoietic cell recovery under stress conditions.

## Materials and methods

### Mice

Mice were maintained under specific pathogen-free conditions in a barrier facility in microisolator cages. This study complied with all ethical regulations involving experiments with mice, and the Institutional Animal Care and Use Committee of Albert Einstein College of Medicine approved all experimental procedures, based on protocol #00001101. C57BL/6J mice (stock #000664), B6.129-Postn[tm2.1(cre/Esr1*)Jmol]/J (*Kanisicak et al., 2016*) mice (stock #029645), *Gt(ROSA)26Sor[tm4(ACTB-tdTomato,-EGFP)Luo]/J* (Rosa26[mT/mG]) mice (stock #007576), and C57BL/6-Tg(UBC-GFP)30Scha/J (stock #004353) mice were ordered from Jackson Laboratory and then bred in our facilities. *Nestin*-GFP (*Méndez-Ferrer et al., 2010*) mice were kindly provided by G. Enikolopov and then bred in our facilities. Cdh5 (VE-cadherin)-CreER; iTdTomato (*Wang et al., 2010*) mice were kindly provided by R. H. Adams (Max Planck Institute for Molecular Biomedicine, Germany) and then bred in our facilities. Unless otherwise specified, 6- to 12-week-old mice were used for the experiments. For all analytical and therapeutic experiments, sex-matched animals from the same age group were randomly assigned to experimental groups.

### Bone transplantation procedure

Donor mice were anesthetized with isoflurane and euthanized by cervical dislocation. Femurs were isolated and preserved in an ice-cold phosphate-buffered saline (PBS) solution with 1% fetal bovine serum (FBS). Recipient mice were anesthetized with a ketamine/xylazine intraperitoneal injection (10 μl/g). Donor femurs were subcutaneously implanted in the back of the recipient mice, and the skin was sutured with a non-absorbable polyamide 5/0 silk. Mice were allowed to recover under a heat lamp until awake and monitored daily for up to a week post-surgery.

### In vivo treatment

For lineage tracing experiments using femurs from Postn[tm2.1(cre/Esr1*)Jmol]/J donor mice, tamoxifen (1 mg/mouse) was administered intraperitoneally to recipient mice twice daily for 10 consecutive days starting at day 2 post-transplantation.

### Bone marrow transplantation

Non-competitive repopulation assays were performed using CD45.1 and CD45.2 mice. Recipient mice were lethally irradiated (12 Gy, two split doses) in a Cesium Mark 1 irradiator (JL Shepherd & Associates). A total of $1 \times 10^6$ CD45.2$^+$ bone marrow nuclear cells from either the graft or host femurs were obtained at 5 months after transplantation and injected retro-orbitally into irradiated CD45.1$^+$ mice. Mice were bled retro-orbitally every 4 weeks after bone marrow transplantation, and peripheral blood was analyzed for engraftment and repopulation up to 16 weeks.

### Preparation of single-cell suspensions

To isolate P-SSCs, muscle tissue was carefully removed using scissors and intact bones were submerged for 30 min in ice-cold PBS with 1% FBS. The periosteum was carefully removed with a surgical blade, and mechanical dissociation was performed using scissors. Enzymatic dissociation was performed by incubating the periosteum fragments for 45 min at 37°C in digestion buffer (Hank's balanced salt solution (Gibco) containing 1 mg ml$^{-1}$ collagenase type IV (Gibco) and 2 mg ml$^{-1}$ dispase (Gibco)) on a rotator. Bone marrow cells were obtained by flushing and dissociating using a 1-ml syringe with PBS via a 21-gauge needle. For analysis of stromal and endothelial cell populations, intact bone marrow plugs were flushed into digestion buffer using 21- or 25-gauge needles and incubated at 37°C for 30 min with manual mixing every 10 min. After bone marrow and periosteum isolation, the remaining compact bone was crushed, mechanically dissociated using scissors as previously described (*Gulati et al., 2018*) and digested in the digestion buffer, rotating for 45 min at 37°C. Enzymatic digestion was stopped by adding ice-cold PEB buffer (PBS with 0.5% BSA and 2 mM EDTA).

## Flow cytometry and cell sorting

For FACS analysis and sorting, red blood cells were lysed (distilled $H_2O$ containing 155 mM ammonium chloride, 10 mM potassium bicarbonate, and 0.5 M EDTA) and washed in ice-cold PEB (PBS containing 0.5% BSA and 2 mM EDTA) before staining with antibodies in PEB for 20 min on ice. Dead cells and debris were excluded by FSC (forward scatter), SSC (side scatter), and DAPI (4′,6-diamino-2-phenylindole; Sigma). FACS analyses were carried out using BD LSRII flow cytometry (BD Biosciences) and cell sorting experiments were performed using a MoFlo Astrios (Beckman Coulter). Data were analyzed with FlowJo 10.4.0 (LCC) and FACS Diva 6.1 software (BD Biosciences). Antibodies used for FACS can be found in *Supplementary file 1*. For metabolic assays, cells were first stained with cell surface markers prior to labeling with metabolic dyes. For cellular ROS quantification, cells were incubated with DHE (5 µM; Molecular Probes) for 20 min at 37°C in PBS. Glucose uptake quantification was performed by incubating the cells in DMEM without glucose (Gibco) containing Glutamax (1:100; Gibco) and 2-NBDG (17 µmol m$^{l-1}$; Cayman Chemical Company) for 30 min at 37°C.

## CFU-F assays

For CFU-F and stromal cell culture, CD45−Ter119−CD31−CD51+CD200+ stromal cells isolated from bone marrow and periosteum were sorted and plated at a clonal density (1000 cell/well) in α-MEM (Gibco) containing 20% FBS (HyClone), 10% MesenCult Stimulatory supplement (StemCell Technologies), and 1% penicillin–streptomycin. Half of the medium was changed at day 7. Cells were cultured for 12–14 days, at the end of which the colonies were scored.

## Osteogenic, adipogenic, and chondrogenic differentiation assays

Trilineage differentiation assays toward the osteogenic, adipogenic, and chondrogenic lineages were performed as previously described (*Pinho et al., 2013*), with minor modifications. Briefly, cells were treated with StemXVivo Osteogenic, Adipogenic, or Chondrogenic mouse differentiation media, according to the manufacturer's instructions (R&D Systems). All cultures were maintained with 5% $CO_2$ in a water-jacketed incubator at 37°C. Osteogenic differentiation was revealed by Alizarin Red S staining. Adipocytes were identified by the typical production of lipid droplets and Bodipy (Invitrogen) staining. Chondrocytes were revealed by Alcian Blue staining.

## Ex vivo culture nutrient deprivation assay

Whole bone marrow from one femur and whole periosteum from two femurs were isolated and digested as previously described and plated in α-MEM (Gibco) containing 20% FBS (HyClone), 1% penicillin–streptomycin, 1% L-glutamine, and βFGF. The medium was changed every 3–4 days. Once a plate reached near confluence, CD45 lineage depletion was performed on both bone marrow and periosteum fractions. Cells were then counted and plated in 12- or 24-well plates at approximately 5000 cells/cm$^2$. Once the plates reached near confluence, media was switched to α-MEM without FBS, 1% L-glutamine, and βFGF. Twelve hours after the medium was switched to 0% FBS medium, the cells were trypsinized, spun down, and stained for cell surface markers. After 4–5 days, flow cytometric apoptosis quantification was performed using the CellEvent Caspase 3/7 kit (Thermo Fisher) following the manufacturer's recommendations.

## Immunofluorescence imaging of bone sections

To stain blood vessels, anti-CD31 and anti-CD144 antibodies were injected intravenously into mice (10 µg, 20 µl of 0.5 µg µl$^{-1}$) and mice were sacrificed for analysis at 10 min after injection. For frozen sections of long bones, femurs and tibias were fixed in 4% paraformaldehyde (PFA) overnight at 4°C. For cryopreservation, the bones were incubated sequentially in 10%, 20%, and 30% sucrose/PBS at 4°C for 1 hr each and embedded and flash frozen in SCEM embedding medium (SECTION-LAB). Frozen sections were prepared at 20 µm thickness with a cryostat (CM3050, Leica) using the Kawamoto's tape transfer method (*Kawamoto and Shimizu, 2000*). For immunofluorescence staining, sections were rinsed with PBS, post-fixed with 4% cold PFA for 10 min, followed by blocking with 20% donkey serum (DS; Sigma) in 0.5% Triton X-100/PBS for 3 hr at room temperature (20–25°C). For perilipin staining, sections were incubated for 1 hr at room temperature in saturation buffer (PBS-DS 10%). The rabbit polyclonal anti-perilipin antibody (clone: D1D8; Cat: 9349; Cell Signaling Technology) was used at 1:100 dilution in 2% DS 0.1% Triton X-100/PBS overnight at 4°C. Periostin staining was performed

using whole-mount femur imaging. The bone marrow was exposed by shaving the bone using a cryo-stat (CM3050, Leica). Shaved femurs were fixed 30 min at 4°C in PBS/PFA 4%. Samples were then incubated in the saturation buffer (PBS-DS 10%) during 1 hr at room temperature. Polyclonal goat anti-periostin antibody (Cat: AF2955; R&D) and monoclonal rat anti-endomucin antibodies (clone: V.7C7; Cat: sc-65495; Santa Cruz) were used at a 1:100 dilution overnight at 4°C in PBS-DS 2%. When necessary, primary antibody staining was followed by three washes with 2% DS 0.1% Triton X-100/PBS and a 30-min incubation with Alexa Fluor 568 or Alexa Fluor 488-conjugated secondary antibodies (Invitrogen) and 0.2% DAPI (Sigma).

## Image acquisition

All images were acquired at room temperature using a Zeiss Axio examiner D1 microscope (Zeiss) with a confocal scanner unit (Yokogawa) and reconstructed in three dimensions with Slide Book software (Intelligent Imaging Innovations). Image analysis was performed using both Slide Book software (Intelligent Imaging Innovations) and the Fiji build of ImageJ (NIH).

## Single limb irradiation

Mice were anesthetized using isoflurane chamber and kept anesthetized throughout the procedure. Mice were irradiated using the Small Animal Radiation Research Platform, SARRP (XStrahl, Surrey, UK). Whole body lead shielding protected the mice except for one hindlimb which was protruding out from under the shield and irradiated to 20 Gy for 319 s in a single fraction. Mice were left under the heat lamp until they recovered.

## RNA isolation and quantitative real-time PCR

mRNA was purified using the Dynabeads mRNA DIRECT Micro Kit (Life technologies – Invitrogen) by directly sorting stromal cells into lysis buffer, and reverse transcription was performed using RNA to cDNA EcoDry Premix (Clontech – Takara Bio) following the manufacturer's instructions. The SYBR green (Roche) method was used for quantitative PCR using the QuantStudio 6 Flex system (Applied Biosystems, Thermo Fisher). All mRNA expression levels were calculated relative to *Gapdh* or *Actb*. *Supplementary file 2* lists the primer sequences used.

## RNA sequencing and analysis

Total RNA from 1000 to 3000 sorted steady BM-MSCs, steady-state P-SSCs, graft P-SSCs, and graft BM-MSCs was extracted using the RNAeasy Plus Micro kit (QIAGEN) and assessed for integrity and purity using an Agilent Bioanalyzer. When applicable, RNA from two mice was combined; however, each replicate contained RNA from distinct mice. RNA-seq data generated from Illumina Novaseq6000 were processed using the following pipeline. Briefly, clean reads were mapped to the mouse genome (GRCm38) using Spliced Transcripts Alignment to a Reference (STAR 2.6.1d). Gene expression levels were calculated and differentially expressed genes were identified using DESeq2 and enriched using clusterProfiler. All RNA sequencing data are available under the SuperSeries dataset GSE222272 in GEO omnibus.

## Statistical analysis

All data are presented as the mean ± SEM. *N* represents the number of mice in each experiment, as detailed in the figure legends. No statistical method was used to predetermine sample sizes; sample sizes were determined by previous experience with similar models of hematopoiesis, as shown in previous experiments performed in our laboratory. Statistical significance was determined by an unpaired, two-tailed Student's *t*-test to compare two groups or a one-way ANOVA with multiple group comparisons. Statistical analyses were performed, and data presented using GraphPad Prism 8 (GraphPad Software), FACS Diva 6.1 software BD Biosciences, FlowJo 10.4.0 (LLC), Slide Book Software 6.0 (Intelligent Imaging Innovations), and QuantStudio 6 Real-Time PCR Software (Applied Biosystem, Thermo Fisher). $*p < 0.05$, $**p < 0.01$, $***p < 0.001$, $****p < 0.0001$.

## Acknowledgements

We would like to thank Colette Prophete and Daqian Sun for technical assistance and Lydia Tesfa and the Einstein Flow Cytometry Core Facility for expert cell sort assistance. We thank Charles Brottier

and Robert Dubin of the Albert Einstein College of Medicine Computational Biology core facility for their help with analysis of RNA sequencing data. This work was supported by the National Institutes of Health (NIH) Grant 5R01DK056638 (to PSF and KG), administrative supplement R01DK056638-23S1 (to KEA), R01DK112976 (to PSF), R56DK130895 (to KG), R01DK130895 (to KG), R01HL162584 (to SP), the NIH training Grant T32GM007288-50 (to KEA), and NYSTEM IIRP C029570A (to PSF). TM was supported by the Fondation ARC pour la Recherche sur le Cancer, the Association pour le Développement de l'Hématologie Oncologie, the Société Française d'Hématologie, the Centre Hospitalier Universitaire de Rennes, and the Philip Foundation. ST was supported by the Japan Society for the Promotion of Science (JSPS) Postdoctoral Fellowship for Research Abroad, the Uehara Memorial Foundation Research Fellowship, and the NYSTEM Empire State Institutional Program in Stem Cell Research. M.M. was supported by the EMBO European Commission FP7 (Marie Curie Actions; EMBO-COFUND2012, GA-2012-600394, ALTF 447-2014), by the New York Stem Cell Foundation (NYSCF) Druckenmiller fellowship, and by the American Society of Hematology (ASH) Research Restart Award. SP was also supported by a Longevity Impetus Grant from Norn Group. This work was also supported by the Albert Einstein Cancer Center core support grant P30CA013330. Experimental figure illustrations were created using BioRender. The content is solely the responsibility of the authors and does not necessarily represent the official views of the National Institutes of Health.

## Additional information

### Funding

| Funder | Grant reference number | Author |
|---|---|---|
| National Institute of Diabetes and Digestive and Kidney Diseases | 5R01DK056638 | Paul Frenette Kira Gritsman |
| National Institute of Diabetes and Digestive and Kidney Diseases | R01DK056638-23S1 | Kemi E Akinnola |
| National Institute of Diabetes and Digestive and Kidney Diseases | R56DK130895 | Kira Gritsman |
| National Institute of Diabetes and Digestive and Kidney Diseases | R01DK130895 | Kira Gritsman |
| National Heart, Lung, and Blood Institute | R01HL162584 | Sandra Pinho |
| National Institute of General Medical Sciences | T32GM007288-50 | Kemi E Akinnola |
| New York State Stem Cell Science | IIRP C029570A | Paul Frenette |
| Fondation ARC pour la Recherche sur le Cancer | | Tony Marchand |
| Association pour le Développement de l'Hématologie Oncologie | | Tony Marchand |
| Société Française d'Hématologie | | Tony Marchand |
| Centre Hospitalier Universitaire de Rennes | | Tony Marchand |
| Philip Foundation | | Tony Marchand |
| Japan Society for the Promotion of Science | | Shoichiro Takeishi |

| Funder | Grant reference number | Author |
|---|---|---|
| Uehara Memorial Foundation | | Shoichiro Takeishi |
| New York State Stem Cell Science | | Shoichiro Takeishi |
| EMBO European Commission FP7 | EMBOCOFUND2012 | Maria Maryanovich |
| New York Stem Cell Foundation | | Maria Maryanovich |
| American Society of Hematology | | Maria Maryanovich |
| Norn Group | | Sandra Pinho |
| NCI | P30CA013330 | Kira Gritsman |
| EMBO European Commission FP7 | Marie Curie Actions | Maria Maryanovich |

The funders had no role in study design, data collection, and interpretation, or the decision to submit the work for publication.

## Author contributions

Tony Marchand, Conceptualization, Data curation, Formal analysis, Supervision, Funding acquisition, Validation, Investigation, Visualization, Methodology, Writing – original draft, Project administration, Writing – review and editing; Kemi E Akinnola, Conceptualization, Data curation, Formal analysis, Funding acquisition, Validation, Investigation, Visualization, Methodology, Writing – original draft, Project administration, Writing – review and editing; Shoichiro Takeishi, Formal analysis, Investigation, Methodology, Writing – original draft; Maria Maryanovich, Conceptualization, Formal analysis, Investigation, Visualization, Writing – original draft; Sandra Pinho, Formal analysis, Investigation, Visualization, Writing – original draft; Julien Saint-Vanne, Software, Formal analysis, Validation, Visualization, Writing – original draft; Alexander Birbrair, Conceptualization, Methodology; Thierry Lamy, Investigation, Visualization; Karin Tarte, Conceptualization, Supervision, Writing – original draft; Paul Frenette, Conceptualization, Resources, Supervision, Funding acquisition, Visualization; Kira Gritsman, Conceptualization, Data curation, Formal analysis, Supervision, Funding acquisition, Validation, Visualization, Writing – original draft, Project administration, Writing – review and editing

## Author ORCIDs

Tony Marchand ⬤ http://orcid.org/0000-0003-4932-2169
Kemi E Akinnola ⬤ http://orcid.org/0000-0003-0746-6762
Sandra Pinho ⬤ https://orcid.org/0000-0002-5241-7364
Kira Gritsman ⬤ https://orcid.org/0000-0002-1367-1167

## Ethics

This study complied with all ethical regulations involving experiments with mice, and the Institutional Animal Care and Use Committee of Albert Einstein College of Medicine approved all experimental procedures, based on IACUC protocol #00001101.

Reviewer #1 (Public review): https://doi.org/10.7554/eLife.101714.3.sa1
Reviewer #2 (Public review): https://doi.org/10.7554/eLife.101714.3.sa2
Author response https://doi.org/10.7554/eLife.101714.3.sa3

# Additional files

## Supplementary files
MDAR checklist

Supplementary file 1. Antibodies used in flow cytometry experiments.

Supplementary file 2. Sequences of primers used for RT-PCR .

## Data availability

All RNA sequencing data are available under the SuperSeries dataset GSE222272 in GEO omnibus.

The following dataset was generated:

| Author(s) | Year | Dataset title | Dataset URL | Database and Identifier |
|---|---|---|---|---|
| Marchand T, Akinnola M | 2023 | Periosteal skeletal stem cells can migrate into the bone marrow and support hematopoiesis after injury | https://www.ncbi. nlm.nih.gov/geo/ query/acc.cgi?acc= GSE222272 | NCBI Gene Expression Omnibus, GSE222272 |

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

# Appendix 1

## Appendix 1—key resources table

| Reagent type (species) or resource | Designation | Source or reference | Identifiers | Additional information |
|---|---|---|---|---|
| Strain, strain background (*M. musculus*) | B6.129S-*Postn*^tm2.1(cre/Esr1*)Jmol^/J | *Kanisicak et al., 2016* | Strain #:029645 RRID:IMSR_JAX:029645 | |
| Strain, strain background (*M. musculus*) | Nestin-GFP | *Méndez-Ferrer et al., 2010* | Strain #:033927 RRID:IMSR_JAX:033927 | |
| Strain, strain background (*M. musculus*) | Gt(ROSA)26Sortm4(ACTB-tdTomato,-EGFP)Luo/J (RosamT/mG) | *Muzumdar et al., 2007* | Strain #:007576 RRID:IMSR_JAX:007576 | |
| Strain, strain background (*M. musculus*) | C57BL/6-Tg(UBC-GFP)30Scha/J | *Liu et al., 2020* | Strain #:004353 RRID:IMSR_JAX:004353 | |
| Strain, strain background (*M. musculus*) | Cdh5-CreER; iTdTomato | H. Adams (Max Planck Institute for Molecular Biomedicine, Germany) | | |
| Strain, strain background (*M. musculus*) | C57BL/6 | The Jackson Laboratory/ Bred in lab | Strain #:000664 RRID:IMSR_JAX:000664 | |
| Antibody | anti-streptavidin FITC/ APC-Cy7 | Invitrogen | RRID:AB_10053373 (FITC), AB_10054651 (APC-Cy7) | FC (1:100) |
| Antibody | anti-CD45 APC-eFluor 780/ Pacific Blue | Invitrogen (APC-eFluor 780), Biolegend (Pacific Blue) | APC- eFluor 780 (RRID:AB_1548781), Pacific Blue (RRID:AB_493535) | Clone: 30-F11 FC (1:100) |
| Antibody | anti-CD11b AF647 | Biolegend | RRID:AB_389327 | Clone: M1/70 FC (1:100) |
| Antibody | anti-CD4 PE-Cy7 | Invitrogen | RRID:AB_469576 | Clone: GK1.5 FC (1:100) |
| Antibody | anti-CD8a PE-Cy7 | Invitrogen | RRID:AB_469584 | Clone: 53-6.7 FC (1:100) |
| Antibody | anti-B220 PerCP-Cy5.5 | Invitrogen | RRID:AB_1107006 | Clone: RA3-6B2 FC (1:100) |
| Antibody | anti-CD45.1 PE | Biolegend | RRID:AB_389327 | Clone: A20 FC (1:100) |
| Antibody | anti-CD45.2 FITC | Biolegend | RRID:AB_313442 | Clone: 104 FC (1:100) |
| Antibody | anti-CD144 AF647 | Biolegend | RRID:AB_10568319 | Clone: BV13 FC (1:100) IF (1:10) |
| Antibody | anti-Gr-1 APC-eFluor 780 | Invitrogen | RRID:AB_2534437 | Clone: RB6-8C5 FC (1:100) |
| Antibody | anti-Ter119 Pacific Blue/ APC-eFlour 780 | Biolegend (Pacific Blue), Invitrogen (APC- eFluor 780) | RRID:AB_1548786 (APC- eFluor 780), AB_225116 (Pacific Blue) | Clone: Ter-119 FC (1:100) |
| Antibody | anti-CD31 PE-Cy7/AF647 | Biolegend | RRID:AB_830756 (PE-Cy7), AB_493410 (AF647) | Clone: 390 FC (1:100) IF (1:10) |
| Antibody | anti-CD51 biotin/ PE | Biolegend (biotin), Invitrogen (PE) | RRID:AB_313072 (biotin), AB_465704 (PE) | Clone: RMV-7 FC (1:100) |

*Appendix 1 Continued on next page*

*Appendix 1 Continued*

| Reagent type (species) or resource | Designation | Source or reference | Identifiers | Additional information |
|---|---|---|---|---|
| Antibody | anti-CD200 APC | Biolegend | RRID:AB_10900447 | Clone: OX-90 FC (1:100) |
| Antibody | Anti-perilipin rabbit polyclonal | Cell Signalling Technology | Catalog #: 9349 | Clone D1D8 IF (1:100) |
| Antibody | Anti-periostin goat polyclonal | R&D | Catalog #: AF2955 | IF (1:100) |
| Antibody | Anti-endomucin rat monocloncal | Santa Cruz | Catalog #: sc-65495 | Clone: V.7C7 (1:100) |
| Chemical compound | Collagenase Type IV | Gibco | Catalog #: 17104019 | |
| Chemical compound | Dispase II | Gibco | Catalog #: 17105041 | |
| Chemical compound | Glutamax | Gibco | Catalog #: 35050061 | (1:100) |
| Chemical compound | 10% MesenCult Stimulatory Supplement | StemCell Technologies | Catalog #: 05402 | |
| Chemical compound | 2-NDBG | Cayman Chemical Company | Catalog #: 11046 | |
| Chemical compound | SCEM embedding media | Section-lab | *Kawamoto and Shimizu, 2000* | |
| Chemical Compound | StemXVivo Osteogenic/ Adipogenic Base media | R&D Systems | Catalog #: CCM007 | |
| Chemical Compound | StemXVivo Chondrogenic Base Media | R&D Systems | Catalog #: CCM005 | |
| Chemical Compound | StemXVivo Osteogenic Supplement | R&D Systems | Catalog #: CCM009 | |
| Chemical Compound | StemXVivo Chondrogenic Supplement | R&D Systems | Catalog #: CCM006 | |
| Chemical Compound | StemXVivo Adipogenic Supplement | R&D Systems | Catalog #: CCM011 | |
| Commercial assay, kit | Bodipy 493/503 | Invitrogen | Catalog #: D3922 | |
| Commercial assay, kit | RNA to cDNA EcoDry Premix | Clontech – Takara Bio | Catalog #: 639543 | |
| Commercial assay, kit | CellEvent Caspase 3/7 Green | Thermo Fisher | Catalog #: C10423 | |
| Commercial assay, kit | RNAeasy Plus Micro Kit | QIAGEN | Catalog #: 74034 | |
| Commercial assay, kit | Dynabeads mRNA DIRECT Micro Kit | Invitrogen | Invitrogen 61021 | |
| Sequence-based reagent | *Cxcl12_F* | This paper | PCR primer | 5'-CGCCAAGGTCGTCGCCG-3' |
| Sequence-based reagent | *Cxcl12_R* | This paper | PCR primer | 5'-TTGGCTCTGGCGATGTGGC-3' |
| Sequence-based reagent | *Kitl_F* | This paper | PCR primer | 5'-CCCTGAAGACTCGGGCCTA-3' |
| Sequence-based reagent | *Kitl_R* | This paper | PCR primer | 5'-CAATTACAAGCGAAATGAGA GCC-3' |
| Sequence-based reagent | *Opn_F* | This paper | PCR primer | 5'-AGCAAGAAACTCTTCCAAGC AA-3' |

*Appendix 1 Continued on next page*

*Appendix 1 Continued*

| Reagent type (species) or resource | Designation | Source or reference | Identifiers | Additional information |
| --- | --- | --- | --- | --- |
| Sequence-based reagent | Opn_R | This paper | PCR primer | 5′-GTGAGATTCGTCAGATTCATCCG-3′ |
| Sequence-based reagent | Angpt1_F | This paper | PCR primer | 5′-CTCGTCAGACATTCATCATCCAG-3′ |
| Sequence-based reagent | Angpt1_R | This paper | PCR primer | 5′-CACCTTCTTTAGTGCAAAGGCT-3′ |
| Sequence-based reagent | Vcam1_F | This paper | PCR primer | 5′-GACCTGTTCCAGCGAGGGTCTA-3′ |
| Sequence-based reagent | Vcam1_R | This paper | PCR primer | 5′-CTTCCATCCTCATAGCAATTAAGGTG-3′ |
| Sequence-based reagent | Cdkn1a_F | This paper | PCR primer | 5′-CCTGGTGATGTCCGACCTG-3′ |
| Sequence-based reagent | Cdkn1a_R | This paper | PCR primer | 5′-CCATGAGCGCATCGCAATC-3′ |
| Sequence-based reagent | Cdkn1c_F | This paper | PCR primer | 5′-CGAGGAGCAGGACGAGAATC-3′ |
| Sequence-based reagent | Cdkn1c_R | This paper | PCR primer | 5′-GAAGAAGTCGTTCGCATTGGC-3′ |
| Sequence-based reagent | Cdk4_F | This paper | PCR primer | 5′-ATGGCTGCCACTCGATATGAA 3′ |
| Sequence-based reagent | Cdk4_R | This paper | PCR primer | 5′-TCCTCCATTAGGAACTCTCACAC-3′ |
| Sequence-based reagent | Sod1_F | This paper | PCR primer | 5′-TGGTGGTCCATGAGAAACAA-3′ |
| Sequence-based reagent | Sod1_R | This paper | PCR primer | 5′-GTTTACTGCGCAATCCCAAT-3′ |
| Sequence-based reagent | Gls_F | This paper | PCR primer | 5′-TTCGCCCTCGGAGATCCTAC-3′ |
| Sequence-based reagent | Gls_R | This paper | PCR primer | 5′-CCAAGCTAGGTAACAGACCCT-3′ |
| Sequence-based reagent | Gpx1_F | This paper | PCR primer | 5′-AGTCCACCGTGTATGCCTTCT-3′ |
| Sequence-based reagent | Gpx1_R | This paper | PCR primer | 5′-GAGACGCGACATTCTCAATGA-3′ |
| Sequence-based reagent | Postn_F | This paper | PCR primer | 5′-CCTGCCCTTATATGCTCTGCT-3′ |
| Sequence-based reagent | Postn_R | This paper | PCR primer | 5′-AAACATGGTCAATAGGCATCACT-3′ |
| Sequence-based reagent | Gapdh_F | This paper | PCR primer | 5′-TGTGTCCGTCGTGGATCTGA-3′ |
| Sequence-based reagent | Gapdh_R | This paper | PCR primer | 5′-CCTGCTTCACCACCTTCTTGA-3′ |
| Sequence-based reagent | Actb_F | This paper | PCR primer | 5′-GCTTCTTTGCAGCTCCTTCGT-3′ |
| Sequence-based reagent | Actb_R | This paper | PCR primer | 5′-ATCGTCATCCATGCCGAACT-3′ |
| Other | DAPI | Sigma | Prod #: D9542 | |

