## [Editor Report · eLife Assessment]

The study presents **valuable** insights into the role of periosteal stem cells in bone marrow regeneration. The evidence is **convincing**. The data broadly support their claims and in line with state-of-art methodology. Future study on their model will help to strengthen their discovery further.

---

## [Referee Report · Reviewer #1 (Public review)]

Summary:

The manuscript under review investigates the role of periosteal stem cells (P-SSC) in bone marrow regeneration using a whole bone subcutaneous transplantation model. While the model is somewhat artificial, the findings were interesting, suggesting the migration of periosteal stem cells into the bone marrow and their potential to become bone marrow stromal cells. This indicates a significant plasticity of P-SSC consistent with previous reports using fracture models (Cell Stem Cell 29:1547, Dev Cell 59:1192).

Major comments from previous round of review:

(1) The authors assert that the periosteal layer was completely removed in their model, which is crucial for their conclusions. To substantiate this claim, it is recommended that the authors provide evidence of the successful removal of the entire periosteal stem cell (P-SSC) population. A colony-forming assay, with and without periosteal removal, could serve as a suitable method to demonstrate this.

(2) The observation that P-SSCs do not express Kitl or Cxcl12, while their bone marrow stromal cell (BM-MSC) derivatives do, is a key finding. To strengthen this conclusion, the authors are encouraged to repeat the experiment using Cxcl12 or Scf reporter alleles. Immunofluorescence staining that confirms the migration of periosteal cells and their transformation into Cxcl12- or Scf-reporter-positive cells would significantly enhance the paper's key conclusion.

(3) On page 8, line 20, the authors' statement regarding the detection of Periostin+ cells outside the periosteum layer could be misinterpreted due to the use of the periostin antibody. Given that periostin is an extracellular matrix protein, the staining may not accurately represent Periostin-expressing cells but rather the presence of periostin in the extracellular matrix. The authors should revise this section for greater precision.

Comments on revised version:

My comments from the previous round of review have mostly been addressed.

---

## [Referee Report · Reviewer #2 (Public review)]

Summary:

The authors have established a femur graft model that allows the study of hematopoietic regeneration following transplantation. They have extensively characterized this model, demonstrating the loss of hematopoietic cells from the donor femur following transplantation, with recovery of hematopoiesis from recipient cells. They also show evidence that BM MSCs present in the graft following transplantation are graft-derived. They have utilized this model to show that following transplantation, periosteal cells respond by first expanding, then giving rise to more periosteal SSCs, then migrating into the marrow to give rise to BM MSCs.

Strengths:

These studies are notable in several ways: (1) establishment of a novel femur graft model for the study of hematopoiesis; (2) Use of lineage tracing and surgery models to demonstrate that periosteal cells can give rise to BM MSCs.

Weaknesses:

There are a few weaknesses. First, the authors do not definitively demonstrate the requirement of periosteal SSC movement into the BM cavity for hematopoietic recovery. Hematopoiesis recovers significantly before 5 months, even before significant P-SSC movement has been shown, and hematopoiesis recovers significantly even when periosteum has been stripped. Second, it is not clear how the periosteum is changing in the grafts. Which cells are expanding is unclear, and it is not clear if these cells have already adopted a more MSC-like phenotype prior to entering the marrow space. Indeed, given the presence of host-derived endothelial cells in the BM, these studies are reminiscent of prior studies from this group and others that re-endothelialization of the marrow may be much more important for determining hematopoietic regeneration, rather the P-SSC migration. Third, the studies exploring the preferential depletion of BM MSCs vs P-SSCs are difficult to interpret. The single metabolic stress condition chosen was not well-justified, and the use of purified cell populations to study response to stress ex vivo may have introduced artifacts into the system.

Comments on the current version: The authors have addressed my concerns adequately

---

## [Author Response]

The following is the authors’ response to the original reviews.

**Public Reviews:**

**Reviewer #1 (Public review):**
The manuscript under review investigates the role of periosteal stem cells (P-SSC) in bone marrow regeneration using a whole-bone subcutaneous transplantation model. While the model is somewhat artificial, the findings were interesting, suggesting the migration of periosteal stem cells into the bone marrow and their potential to become bone marrow stromal cells. This indicates a significant plasticity of P-SSC consistent with previous reports using fracture models (Cell Stem Cell 29:1547, Dev Cell 59:1192).Major Concerns(1) The authors assert that the periosteal layer was completely removed in their model, which is crucial for their conclusions. To substantiate this claim, it is recommended that the authors provide evidence of the successful removal of the entire periosteal stem cell (P-SSC) population. A colony-forming assay, with and without periosteal removal, could serve as a suitable method to demonstrate this.

We are grateful to the reviewer for this valuable suggestion. The objective of this experiment was to demonstrate that periosteal ablation impairs bone marrow regeneration, a finding that is supported by our results. We expect that ablation of the periosteum would be associated with only a partial decrease in CFU-F activity, given the presence of MSCs in the bone and in the endosteal region of the bone marrow. Therefore, CFU-F assays would be difficult to interpret in this setting. In view of the phenotype obtained, providing proof of concept of the importance of the periosteum, we do not believe that further experiments would strengthen the level of proof of this experiment.

(2) The observation that P-SSCs do not express Kitl or Cxcl12, while their bone marrow stromal cell (BM-MSC) derivatives do, is a key finding. To strengthen this conclusion, the authors are encouraged to repeat the experiment using Cxcl12 or Scf reporter alleles. Immunofluorescence staining that confirms the migration of periosteal cells and their transformation into Cxcl12- or Scf-reporter-positive cells would significantly enhance the paper's key conclusion.

Transplantation of periosteum isolated from Cxcl12 or Scf into WT bones is an excellent suggestion. Indeed, this experiment would confirm (1) the migration of periosteal SSC and (2) the expression of Cxcl12 and Scf by BM-MSCs derived from the periosteum .However, it should be noted that the current limitations in terms of available resources preclude the execution of these experiments. Moreover, the use of the PostnCre^ER^;Tmt mice represent the optimal approach for tracking and specifically isolating BM-MSCs derived from the periosteum. The expression of Cxcl12 and Scf by BM-MSCs derived from the periosteum has been demonstrated in 2 distinct experimental models (Figures 5 and 6).

(3) On page 8, line 20, the authors' statement regarding the detection of Periostin+ cells outside the periosteum layer could be misinterpreted due to the use of the periostin antibody. Given that periostin is an extracellular matrix protein, the staining may not accurately represent Periostin-expressing cells but rather the presence of periostin in the extracellular matrix. The authors should revise this section for greater precision.

We acknowledge and appreciate the reviewer's attention to detail. This is, in fact, an error. Nestin-GFP positive periosteal SSC are seen within the periosteum marked by an anti-periostin antibody labeling the extracellular matrix of the periosteum. The manuscript has been revised to address this inaccuracy on page 9, lines 8-9.

**Reviewer #2 (Public review):**
Summary:The authors have established a femur graft model that allows the study of hematopoietic regeneration following transplantation. They have extensively characterized this model, demonstrating the loss of hematopoietic cells from the donor femur following transplantation, with recovery of hematopoiesis from recipient cells. They also show evidence that BM MSCs present in the graft following transplantation are graft-derived. They have utilized this model to show that following transplantation, periosteal cells respond by first expanding, then giving rise to more periosteal SSCs, and then migrating into the marrow to give rise to BM MSCs.Strengths:These studies are notable in several ways:(1) Establishment of a novel femur graft model for the study of hematopoiesis;(2) Use of lineage tracing and surgery models to demonstrate that periosteal cells can give rise to BM MSCs.

We thank the reviewer for noting the novelty of our manuscript.

Weaknesses:There are a few weaknesses. First, the authors do not definitively demonstrate the requirement of periosteal SSC movement into the BM cavity for hematopoietic recovery. Hematopoiesis recovers significantly before 5 months, even before significant P-SSC movement has been shown, and hematopoiesis recovers significantly even when periosteum has been stripped.This is an important point. Notably, we can see expansion of P-SSCs by day 8 after femur transplantation and evidence of periosteum-derived SSCs in the bone marrow by day 15, before we can detect any significant hematopoietic recovery (see Figure 3A-C).Second, it is not clear how the periosteum is changing in the grafts. Which cells are expanding is unclear, and it is not clear if these cells have already adopted a more MSC-like phenotype prior to entering the marrow space.This is an interesting question. To examine early changes in gene expression in periosteal SSCs in grafted femurs, we performed additional RNA sequencing on host periosteal SSCs vs periosteal SSCs from grafted femurs at an earlier time point - at 3 days after femur transplantation and on host bone marrow MSCs (see new Supplementary Figure S5 A-C). At this time point the three cell populations are already distinct on the PCA plot (Figure S5A), and there is downregulation of some periosteal genes in the graft P-SSCs (Figure S5B). However, we do not yet see upregulation of Kitl or Cxcl12 or most other BM MSC genes in graft P-SSCs at this time point (Figure S5B). Furthermore, gene set enrichment analysis (GSEA) revealed upregulation of cell cycle, DNA replication and mismatch repair gene signatures, and downregulation of multiple gene signatures compared to host P-SSCs (Figure S5C). Therefore, we conclude that P-SSCs already adopt some gene expression changes early after femur transplantation, but have not yet fully differentiated into BM MSCs at this early time point. This experiment is now discussed on p.10 of the revised manuscript.Indeed, given the presence of host-derived endothelial cells in the BM, these studies are reminiscent of prior studies from this group and others that re-endothelialization of the marrow may be much more important for determining hematopoietic regeneration, rather than the P-SSC migration.

Indeed, as previously shown by our group and others, we agree that endothelial regeneration and re-endothelialization may also play an important role in this bone marrow regeneration model. It is noteworthy that this model has the potential to serve as a valuable tool for analyzing the origin of BM endothelial cells during regeneration processes. To further illustrate the endothelial regeneration, additional images of bone sections from VE-cadherin-cre;TdTomato grafted femurs at 15 days, one month, and five months post-transplantation have been included in the new Figure S3. These images reveal extensive vascularization of the graft and proximity of UBC-GFP+ donor-derived vessels to VE-cadherin+ host-derived blood vessels in the bone marrow within one month (see Figure S2C). This observation is consistent with the timing of both BM MSC recovery and HSC recovery in the grafts, thereby suggesting the importance of endothelial recovery (see Fig. 1B). A new discussion of these findings has been included on page 6 of the revised manuscript and on page 16 in the discussion section.

Third, the studies exploring the preferential depletion of BM MSCs vs P-SSCs are difficult to interpret. The single metabolic stress condition chosen was not well-justified, and the use of purified cell populations to study response to stress ex vivo may have introduced artifacts into the system.

We chose to focus on hypoxia as the main condition in which to analyze the stress response of P-SSCs vs BM MSCs because we reasoned that due to the location of P-SSCs on the outside of the bone, these cells would be exposed to a higher oxygen tension than BM-MSCs, which are located within the bone marrow. Therefore, we wanted to determine whether this exposure to a different oxygen tension would be sufficient to explain the different properties of P-SSCs and BM MSCs. We modified the text on p.11 of the manuscript to explain the rationale for this experiment better.

**Reviewer #3 (Public review):**
Summary:Marchand, Akinnola, et al. describe the use of the novel model to study BM regeneration. Here, they harvest intact femurs and subcutaneously graft them into recipient mice. Similar to standard BM regeneration models, there is a rapid decrease in cellularity followed by a gradual recovery over 5 months within the grafts. At 5 months, these grafts have robust HSC activity, similar to HSCs isolated from the host femur. They find that periosteum skeletal stem cells (p-SSCs) are the primary source of BM-MSCs within the grafted femur and that these cells are more resistant to the acute stress of grafting the femur.Strengths:This is an interesting manuscript that describes a novel model to study BM regeneration. The model has tremendous promise.

We thank the reviewer for highlighting the novelty and potential of our work.

Weaknesses:The authors claim that grafting intact femurs subcutaneously is a model of BM regeneration and can be used as a replacement for gold standard BM regeneration assays such as sublethal chemo/irradiation. However, there isn't enough explanation as to how this model is equivalent or superior to the traditional models. For instance, the authors claim that this model allows for the study of "BM regeneration in vivo in response to acute injury using genetic tools." This can and has been done numerous times with established, physiologically relevant BM regeneration models. The onus is on the authors to discuss or perform the necessary experiments to justify the use of this model. For example, standard BM regeneration models involve systemic damage that is akin to therapies that require BM regeneration. How is studying the current model that provides only an acute injury more relevant and useful than other models? As it stands, it seems as if the authors could have done all the experiments demonstrating the importance of these p-SSCs in the traditional myelosuppressive BM regeneration models to be more physiologically relevant. Along these lines, the use of a standard BM regeneration model (e.g., sublethal chemo/irradiation) as a critical control is missing and should be included. Even if the control doesn't demonstrate that p-SSCs can contribute to the BM-MSC during regeneration, it will still be important because it could be the justification for using the described model to specifically study p-SSCs' regulation of BM regeneration.We appreciate the reviewer raising this important point. We never intended this femur transplantation model of bone marrow injury to replace more established models, such as chemotherapy or irradiation. In fact, we compared the effects of femur transplantation to localized bone irradiation on P-SSCs using our Periostin-Cre;Td-Tomato lineage tracing model. We found that irradiation does not induce the same migration of Tomato+ P-SSCs from the periosteum to the bone marrow cavity the way that femur transplantation, and cannot be used to demonstrate the plasticity of P-SSCs in the same way (see new Supplementary Figure S7D-E). Therefore, this appears to be a more severe form of bone marrow injury, and is not similar to other more established assays of bone marrow injury. We also added this discussion to the revised manuscript on p.14 and in the discussion section on p.17.The authors perform some analysis that suggests that grafting a whole femur mimics BM regeneration, but there are many experiments missing from the manuscript that will be necessary to support the use of this model. To demonstrate that this new model mimics current BM regeneration models, the authors need to perform a careful examination of the early kinetics of hematopoietic recovery post-transplant. Complete blood counts should be performed on the grafts, focusing on white blood cells (particularly neutrophils), red blood cells, platelets, all critical indicators of BM regeneration. This analysis should be done at early time points that include weekly analysis for a minimum of 28 days following the graft. Additionally, understanding how and when the vasculature recovers is critical. This is particularly important because it is well-established that if there is a delay in vascular recovery, there is a delay in hematopoietic recovery. As mentioned above, a standard BM regeneration model should be used as a control.

We concur with the reviewer that hematopoietic recovery is a pivotal aspect of this model. We conducted a time-course analysis of bone marrow and HSC cellularity from day 0 to month 5 post-transplantation (Figure 1B). Furthermore, we evaluated the HSC capacities through bone marrow transplantation from grafted or host femurs (Figures 1D and 1E) and quantified the various hematopoietic cells in the graft after five months (Supplemental Figure 1). Furthermore, hematopoiesis occurring in the transplanted bone was comprehensively evaluated in another article, currently in revision and available in BioRxiv (Takeishi, S., Marchand, T., Koba, W. R., Borger, D. K., Xu, C., Guha, C., Bergman, A., Frenette, P. S., Gritsman, K., & Steidl, U. (2023). Haematopoietic stem cell numbers are not solely determined by niche availability. bioRxiv: the preprint server for biology, 2023.10.28.564559. https://doi.org/10.1101/2023.10.28.564559). We did not use another assay of bone marrow regeneration as a “control”, since we do not expect to see similar plasticity of periosteal SSCs in these models, such as with the localized irradiation model described in the new Figure S7D-E.

We agree with the reviewer that endothelial recovery is also likely to be very important for hematopoietic recovery in this model, but this was not the focus of this manuscript. The process of endothelial recovery is likely to be more complex than that of MSC recovery, as our findings indicate that the graft endothelium can arise from both the host and the graft femur (see Fig.2D). Consequently, further investigation into the mechanisms of endothelial recovery and its contribution to hematopoiesis in this experimental system will be an interesting focus of future work. We believe that this bone transplantation model represents a valuable tool for addressing questions regarding the origin and regeneration mechanisms of bone marrow endothelial cells.

The contribution of donor and host cells to the BM regeneration of the graft is interesting. Particularly, the chimerism of the vasculature. One can assume that for the graft to undergo BM regeneration, there needs to be the delivery of nutrients into the graft via the vasculature. The chimerism of the vascular network suggests that host endothelial cells anastomose with the graft. Host mice should have their vascular system labeled with a dye such as dextran to determine if anastomosis has occurred. If not, the authors need to explain how this graft survives up to 5 months. If anastomosis does occur, then it is very surprising that the hematopoietic system of the graft is not a chimera because this would essentially be a parabiosis model. This needs to be explained.

We have included additional images of bone sections from VE-cadherin-cre;tdTomato grafted femurs at 15 days, one month, and five months post transplantation in the new Figure S3. These images show extensive vascularization of the graft and proximity of UBC-GFP+ donor-derived vessels to VE-cadherin+ host-derived blood vessels in the bone marrow within one month, suggesting a potential anastomosis (Figure S2C). However, it is not surprising that hematopoiesis arises exclusively from the host, as we observed complete death of the hematopoietic cells and BM MSCs in the graft femur within the first 3 days of femur transplantation (see Figure S1A), and we do not see any significant hematopoietic recovery in the grafts until at least 2 months (see Fig.1B). Therefore, this is not similar to a parabiosis model, as confirmed by our chimerism studies shown in Figure 2D. In addition, these data are consistent with the results reported with the use of ossicles (doi:10.1038/nature09262; DOI 10.1016/j.cell.2007.08.025; doi:10.1038/nature07547).

Most of the data presented for the resistance of p-SSCs to stress suggests DNA damage response. Do p-SSCs demonstrate a higher ability to resolve DNA damage? Do they accumulate less DNA damage? Staining for DNA damage foci or performing comet assays could be done to further define the mechanism of stress resistance properties of p-SSCs.

This is an interesting question. In our RNA sequencing analysis of graft P-SSCs compared with host P-SSCs we did observe an upregulation of mismatch repair gene signatures by gene set enrichment analysis (GSEA) (new Figure S5C). Therefore, it is possible that P-SSCs do have an altered DNA damage response. However, we are unable to investigate this further at this time.

Given the importance of BM-MSCs in hematopoiesis and that the majority of the emerging BM-MSCs appear to be derived from p-SSCs, the authors should perform experiments to determine if p-SSC-derived BM-MSCs are critical regulators of BM regeneration. For example, the authors could test this by crossing the Postn-creER mice with iDTR mice to ablate these cells and see if recovery is inhibited or delayed. This should be done with the described periosteum-wrapped femur graft model as well as a control BM regeneration model. Demonstrating that the deletion of these cells affects BM regeneration in both models would further justify the physiological relevance and utility of the femur graft model.

We thank the reviewer for this excellent suggestion, and we agree that this is an important experiment. However, our attempts to ablate Postn+ cells using the iDTA system were limited by technical difficulties, which we are unable to address at this time.

**Recommendations for the authors:**

**Reviewer #1 (Recommendations for the authors):**
(1) In Figure 2C, the vascular network staining appears to be duplicated, suggesting a possible error in image capture. The authors should replace this image with a different field or an alternative picture to avoid confusion.

We thank the reviewer for noting this accidental duplication due to an image stitching problem. Figure 2C was replaced by a different image from the same experiment.

(2) For consistency and clarity, a scale bar should be included in Figure S3E to indicate that the magnification factors of the respective visual fields are identical.

We thank the reviewer for highlighting this point. The magnification used has been added in the revised Figure.

(3) In Figure S5B, the difference in normalized Opn mRNA expression relative to Gapdh between steady-state BM-MSCs and P-SSCs seems substantial, which contradicts the "ns" (not significant) label. The authors should verify the accuracy of this labeling.

We agree with the reviewer that this difference in what is now Figure S6B looks substantial. However, we confirmed that this difference is not statistically significant, likely due to the high variability between replicates in Opn expression in the steady state BM MSCs.

**Reviewer #2 (Recommendations for the authors):**
In order to strengthen the argument that P-SSCs are necessary for hematopoietic recovery, the authors should consider providing the following data:(1) In the periosteal stripping experiments, the authors should show if periosteum-derived MSCs are present in the BM throughout the process of hematopoietic recovery (not just at the end of the experiment). If none are present at the end, that would mean that periosteum is not required for hematopoietic recovery, but would still suggest that it is required for optimal hematopoietic recovery. At early time points, it would also be very helpful to demonstrate the composition and amount of endothelium present in the marrow to determine if P-SSC migration and differentiation into MSCs depends on endothelial reconstitution.

To further examine the vascularization of the transplanted femur at an earlier time point, we have added additional images of grafted femur from VE-cadherin-cre;tdTomato at 15 days and one month post transplantation in the new Figure S3A and S3B. These images already show extensive vascularization of the graft periosteum stained with an anti-periostin antibody. In addition, we observed anastomoses of host VE-cadherin;Tmt+ blood vessels with graft ubc-GFP+ blood vessels in the grafted periosteum within one month (Figure S3C).

(2) Studies of the surgical periosteum grafts could benefit from histologic analysis of the BM and its MSC components at earlier time points following grafting since the data provided are only at 5 months. Such studies would allow a better appreciation of the relationship between P-SSC migration into the marrow and hematopoietic recovery.

We have performed histologic analysis of grafted femurs at multiple early time points, which shows expansion of P-SSCs and their migration into the bone marrow cavity (Figure 3C).

(3) Studies of stress responses preferably should be performed using intact bone and should characterize P-SSC and BM MSC apoptosis, cell cycle status, differentiation, etc, immediately following shifts to the stress conditions. These studies would be more compelling if performed using additional "stress" conditions likely to represent the graft environment.

This is an interesting suggestion. However, these types of studies would not be possible in intact bones ex vivo, as P-SSCs are known to migrate out of the bone in culture.